



# Molecular and spatial distributions of dicarboxylic acids, oxocarboxylic acids and α-dicarbonyls in marine aerosols over the South China Sea to East Indian Ocean

Jing Yang[1,2], Wanyu Zhao[2,4], Lianfang Wei[2], Qiang Zhang[1], Yue Zhao[1], Wei Hu[1], Libin Wu[1], Xiaodong Li[1], Chandra Mouli Pavuluri[1], Xiaole Pan[2], Yele Sun[2], Zifa Wang[2], Cong-Qiang Liu[1], Kimitaka Kawamura[3], and Pingqing Fu[1]

[1]Institute of Surface-Earth System Science, Tianjin University, Tianjin 300072, China

[2]State Key Laboratory of Atmospheric Boundary Layer Physics and Atmospheric Chemistry, Institute of Atmospheric Physics, Chinese Academy of Sciences, Beijing 100029, China

[3]Chubu Institute for Advanced Studies, Chubu University, Kasugai 487-8501, Japan

[4]College of Earth and Planetary Sciences, University of Chinese Academy of Sciences, Beijing, 100049, China

*Correspondence to*: Pingqing Fu (fupingqing@tju.edu.cn)

**Abstract.** Marine aerosol samples collected from the South China Sea (SCS) to East Indian Ocean (EIO) during a cruise from March 10 to April 26, 2015 were studied for diacids and related compounds. In view of the air mass backward trajectories and source regions of geographical features, the cruise area is segregated into the South China Sea (SCS), the East Indian Ocean off the coast of western Indonesia (EIO-WI), the EIO off the coast of Sri Lanka (EIO-SL), Malacca and Sri Lanka docking point (SLDP). Total concentrations of diacids, oxoacids and α-dicarbonyls were much higher at SLDP followed by the SCS, Malacca, and the lowest at the EIO-WI. In this study, oxalic acid ($C_2$) is the dominant diacid during the cruise, followed by malonic acid ($C_3$) in the SCS, EIO-WI, EIO-SL and Malacca, whereas succinic acid ($C_4$) diacid was relatively more abundant than $C_3$ diacid in SLDP. Except for SLDP, $C_3/C_4$ mass ratios were always greater than 1, and no significant difference was observed among the cruise. The $C_2/C_4$ and $C_2$/total diacids ratios also showed the similar trends. Average mass ratios of adipic acid ($C_6$) to azelaic acid ($C_9$) were less than unity except for at EIO-WI; the mass ratios of phthalic acid (Ph) to azelaic acid ($C_9$) were less than 2 except for at SCS. The concentrations of diacids were higher when the air masses originated from the terrestrial regions than those from the remote oceanic regions. Based on the molecular distributions of organic acids, the mass ratios and linear correlations of selected compounds in each area, we found that the oxidation of biogenic volatile organic compounds (BVOCs) released from the ocean surface and subsequent photochemical oxidations were the main contributors to diacids, oxocarboxylic acids and α-dicarbonyls over the SCS to EIO. In addition, the continental outflow that enriched with the anthropogenic VOCs and their aging influenced the organic aerosol loading, particularly over the SCS. The emissions from local terrestrial vegetations



as well as fossil fuel combustion and subsequent *in-situ* photochemical oxidation also played a prominent role in controlling the organic aerosols loading and molecular distributions of diacids and related compounds at SLDP.

## 1 Introduction

Atmospheric aerosols containing inorganic and organic materials play vital roles in air quality, atmospheric chemistry and global biogeochemical cycles, and have a critical impact on the global climate system (Ramanathan et al., 2001). Organic aerosols in tropospheric aerosols account for up to 90% (Kanakidou et al., 2005), and most of them are water-soluble (Kanakidou et al., 2005;De Gouw and Jimenez, 2009). The abundant diacids, oxoacids and α-dicarbonyls are reported to be the major fraction of water-soluble organic aerosols (Sorooshian et al.,

2010;Ervens et al., 2011;Cong et al., 2015), which enhance the capacity of aerosol particles to act as cloud condensation nuclei and ice nuclei in the atmosphere (Ervens et al., 2011;Zhao et al., 2016;Vergara-Temprado et al., 2017;Huang et al., 2018), whereas the water-insoluble substances such as lipid class compounds can reduce their hygroscopic activity (Kawamura et al., 2017).

It has been well established that the aerosols in the ocean's atmosphere are influenced by the marine emissions, such as plankton activities (Cavalli et al., 2004;O'Dowd and De Leeuw, 2007;Facchini et al., 2008). Marine

emissions account for more than half of the global natural aerosol burden (1000–3000 Tg yr$^{-1}$) (Iii and Duce, 1988), and thus significantly influence the Earth's climate system (Haywood and M., 1999). Most of the previous studies on marine aerosols have focused mainly on the measurements of non-sea-salt/sea-salt sulfates (Charlson et al., 1987;Charlson et al., 1992) and mineral dust (Schulz et al., 2012). However, diacids and related compounds in

marine aerosols have been paid little attention to, despite the fact that they account for >10% of total carbon (TC) in the remote oceanic regions (Kawamura and Sakaguchi, 1999;Wang and Kawamura, 2006), which highlights their importance in the marine atmosphere.

Generally, molecular distributions of dicarboxylic acids in atmospheric aerosols are characterized by the dominance of oxalic ($C_2$) acid, followed by malonic ($C_3$) and succinic ($C_4$) acids (Fu et al., 2013;Kawamura and

Bikkina, 2016). They can be directly emitted from anthropogenic emissions such as fossil fuel combustion (Kawamura and Kaplan, 1987) and biomass burning (Narukawa et al., 1999;Kawamura et al., 2013), and formed by atmospheric oxidation of various volatile organic compounds (VOCs) emitted from primary sources (Kawamura et al., 1996b). In addition, unsaturated fatty acids emitted from the oceanic biota can be further photo-oxidized to form the corresponding diacids in the atmosphere (Kawamura and Sakaguchi, 1999;Rinaldi et al., 2011). Isoprene,



an abundant VOC from terrestrial higher plants, is a major precursor of secondary organic aerosols (Lim et al., 2005;Carlton et al., 2009), including low molecular weight diacids ($C_2$–$C_4$) (Nguyen et al., 2010).

  Chemical characterization of dicarboxylic acids, oxoacids and α-dicarbonyls in atmospheric aerosols provides deeper knowledge on the relative contribution of possible primary sources, the long-range atmospheric transport and the photo-oxidation routes of organic compounds (Kawamura and Bikkina, 2016). Many studies have

investigated the impacts of the continental air outflows on concentrations, compositions and distributions of organic aerosols in the marine areas (Fu et al., 2013). For example, a long-term observation of aerosols at the Chichi-jima Island in the western North Pacific found a remarkable increase in concentrations of dicarboxylic acids in winter and spring due to the influence of the East Asian air outflow transported by the westerlies (Mochida et al., 2003a). Moreover, organic compounds from marine sources, such as unsaturated fatty acids, are main contributors to the

secondary formation of diacids and related compounds in the ocean atmosphere (Miyazaki et al., 2010;Kawamura and Gagosian, 1987). Bikkina et al. (2014) found that the abundance of $C_2$–$C_6$ and $C_9$ diacids continued to increase with the increase in phytoplankton activities in the North Pacific Ocean. Meanwhile, the oxidation products of isoprene (e.g., pyruvic and glyoxylic acids) observed in the marine boundary layer had a similar concentration trend with that of $C_2$, suggesting that the secondary oxidation of biogenic VOCs from marine sources significantly

contribute to the atmospheric loading of diacids and related compounds.

  Although great progress has been made in the research field of atmospheric organic aerosols, the studies on diacids, oxoacids and α-dicarbonyls in the marine atmosphere are scarce and limited to the Mediterranean, North Pacific, South Pacific, Caribbean and Atlantic Ocean (Kawamura and Bikkina, 2016), while little is known about the molecular composition of marine aerosols over the Indian Ocean. In this study, total suspended particle (TSP)

samples collected on day- and night-time basis over the South China Sea and East Indian Ocean were studied for diacids, oxoacids, and α-dicarbonyls, which provide an ideal opportunity to investigate the spatial distributions of marine organic aerosols that are influenced by organics from both marine emissions and long-range transported continental aerosols. In addition, carbonaceous components, i.e., organic (OC) and elemental carbon (EC) and water-soluble OC (WSOC), and inorganic ions in the TSP samples were measured. Based on molecular and spatial

distributions, and mass ratios of organic compounds and relations with bulk components together with the backward air mass trajectories, we discuss their sources and possible formation pathways in the studied region.



## 2 Materials and methods

### 2.1 Marine aerosol sampling

Marine aerosol sampling was performed on the *R/V* "Shiyan1" during the cruise NORC2015-10 in the South China Sea and the East Indian Ocean (Figure 1) from 10 March to 26 April 2015. TSP samples were collected on daytime ($n = 44$) and nighttime ($n = 43$) using a high-volume air sampler (Kimoto AS 810A, KIMOTO, Japan) placed on the front upper deck of the vessel's navigation room. The exhaust outlet of the ship was located at the stern, and the aerosol samples were collected underway to avoid the potential pollution from the ship emissions. The three

samples collected at the Sri Lanka docking point (SLDP) when the ship was docked in the port, which were seriously polluted by ship exhaust emission. Therefore, these three samples were discussed separately to compare with other samples. The air sampler was operated at a flow rate of 1.0 m$^3$ min$^{-1}$ with pre-combusted (450°C, 6h) quartz filters (25 cm × 20 cm, PALLFLEX®TM, 2500 QAT-UP). We also collected field blanks ($n = 4$) during the cruise. After sampling, the filter samples were wrapped in aluminum foil and packed in zip-lock bag and stored in

dark at –20°C until the chemical analysis.

Five-day air mass backward trajectories arriving at 100 m above the sea level on each day were computed using the HYSPLIT model (https://www.ready.noaa.gov/HYSPLIT.php) (Figure 1). The air masses arriving over the South China Sea (SCS) originated from East Asia and surrounding oceanic regions. Those arrived over the East Indian Ocean off the coast of western Indonesia (EIO-WI) originated from the remote EIO, while those arriving

over the EIO off the coast of Sri Lanka (EIO-SL) were mixed from the remote EIO and Bay of Bengal. The air masses arrived over Malacca from the Southeast Asian subcontinent and the surrounding EIO. In addition, the vessel was docked for 3 days at Sri Lanka docking point (SLDP), during which the air masses were originated from the surrounding oceanic regions. Thus, the TSP samples collected in these different locales during the cruise may have been affected by the long-distance atmospheric transport, and hence we segregated the TSP samples into five

categories: SCS, EIO-WI, EIO-SL, SLDP and Malacca (Figure 1).

### 2.2 Determination of diacids and related compounds

Aerosol samples were determined for water-soluble organic acids using a previously reported method (Kawamura, 1993). Briefly, small pieces (47 mm in diameter) of TSP filter samples were extracted with organic-free Milli-Q water (10 mL) under ultrasonication for three times. Then, the extracts were concentrated to dryness and reacted

with 14% BF$_3$/$n$-butanol at 100°C for 1 hour. The derivatized acids and carbonyls were extracted with $n$-hexane and injected into a split/splitless gas chromatography (GC-FID, Agilent 6980) equipped with an HP-5 column for

the determination of diacids and related compounds. The field blank filters were also analyzed using the same experiment procedure. Recoveries of major organic acids were better than 85%. The analytical errors in duplicate analysis were within 10% for major species. Concentrations of diacids and related compounds were corrected

according to the field blanks.

**2.3 Measurements of WSOC, OC and EC**

For WSOC measurement, 17.3 $cm^2$ or 34.7 $cm^2$ of each filter was extracted with Milli-Q water (20 mL) under ultrasonication for 20 min. The extracts were analyzed using a total organic carbon (TOC) analyzer (TOC-L, 5000A, Shimadzu, Japan) for WSOC. OC and EC were determined using a carbon analyzer (Sunset Laboratory Inc., USA)

using a piece (1.5 $cm^2$) of each filter following the Interagency Monitoring of Protected Visual Environments (IMPROVE) protocol (Pavuluri et al., 2011). The limits of detection (LODs) for both OC and EC were 0.1 μgC $cm^{-2}$ with a precision of >10%. Concentrations of all the carbonaceous components reported here were corrected using the field blanks.

Table 1 shows the concentration levels of OC and EC in TSP samples collected from different marine regions. The

concentrations of EC (4.24±3.02 μg $m^{-3}$) in SLDP aerosols were much higher than those in other regions (SCS: 0.27±0.15 μg $m^{-3}$; EIO-WI: 0.05±0.05 μg $m^{-3}$; EIO-SL: 0.04±0.18 μg $m^{-3}$; and Malacca: 0.24±0.18 μg $m^{-3}$), which proved that the samples of SLDP were seriously polluted by ship exhaust, but other sea areas should be minor.

**3 Results and discussion**

**3.1 Concentrations and molecular distributions**

**3.1.1 Dicarboxylic acids**

The concentrations of dicarboxylic acids, oxocarboxylic acids and α-dicarbonyls in each marine region (SCS, EIO-WI, EIO-SL, SLDP and Malacca) are summarized in Table 1. Their chemical structures are shown in Fig. 2. Most of the organic species were found to be more abundant at SLDP followed by the SCS, Malacca, EIO-WI and EIO-

SL, respectively (Table 1). $C_2$ diacid showed the highest abundance followed by $C_3 > C_4 >$ MeGly $> \omega C_2 > C_5 >$ Ph ≈ Gly $> C_9$ (see Table 1 for full form of abbreviation) among the measured major species over the SCS (Fig. 3a), while $C_3 >$ MeGly $> C_4 > F >$ Ph $> C_6 \approx C_9$ over the EIO-WI (Fig. 3b), $C_3 > C_4 > C_9 \approx$ MeGly $> \omega C_2 >$ Gly $>$ Ph at EIO-SL. The order in Malacca was $C_2 > C_3 > C_4 > kC_7 > C_5 > \omega C_2 > C_9 \approx$ MeGly.



Total concentrations of diacids over Malacca varied from 21.9 to 501 ng m$^{-3}$ (average 207±158 ng m$^{-3}$) and the

SCS from 44.6 to 759 ng m$^{-3}$ (305±186 ng m$^{-3}$), which were similar to those reported over the East China Sea (325

ng m$^{-3}$) and Indian Ocean (301 ng m$^{-3}$) during a round-the-world cruise (Fu et al., 2013). The average concentration

of diacids over the EIO-SL (77.6±73.1 ng m$^{-3}$) was similar with those over the western North Pacific (87 ng m$^{-3}$ in

more biologically influenced aerosols (MBA)) (Miyazaki et al., 2010) and Atlantic Ocean (95 ng m$^{-3}$) (Fu et al.,

2013). Total concentrations of diacids over the EIO-WI varied from 8.28 to 96.3 ng m$^{-3}$ (26.1±23.3 ng m$^{-3}$), which

were comparable to those from the western North Pacific (34 ng m$^{-3}$ in LBA) (Miyazaki et al., 2010) and western

Pacific Ocean (60±39 ng m$^{-3}$) (Wang et al., 2006), and were higher than that from the Southern Ocean (4.5±4.0 ng

m$^{-3}$) (Wang et al., 2006). The average concentration of diacids over SLDP (514±231 ng m$^{-3}$) was lower than those

reported from Gosan, Jeju Island, South Korea (660 ng m$^{-3}$ in 2004 and 636 ng m$^{-3}$ in 2010b) (Kawamura et al.,

2004;Kundu et al., 2010b), the East China Sea (850 ng m$^{-3}$) (Mochida et al., 2003b), California coast (424 ng m$^{-3}$)

(Fu et al., 2013), the urban center of Xi'an, China in different dust events (932-2240 ng m$^{-3}$) (Wang et al., 2015),

the megacity Chennai, India (694.5±176.3 and 640.6±150.6 ng m$^{-3}$ in early and late winter) (Pavuluri et al., 2010),

and was comparable to those from the megacity Chennai, India in summer (502.9±117.9 ng m$^{-3}$) (Pavuluri et al.,

2010). However, the concentration of diacids over SLDP was much higher than that from Chichi-jima Island, Japan

(139 ng m$^{-3}$) (Mochida et al., 2003a). The pattern of diacids abundance order ($C_2 > C_9 > C_4 > C_3 > Ph > C_5$) observed

at SLDP was similar to that in continental aerosols (Kawamura and Bikkina, 2016).

$C_2$ diacid concentrations ranged from 4.43 to 70.1 ng m$^{-3}$ with an average of 18.2 ± 18.4 ng m$^{-3}$ in the EIO-WI,

5.21–232 ng m$^{-3}$ (average 58.2 ± 55.3 ng m$^{-3}$) in the EIO-SL, 14.5–373 ng m$^{-3}$ (145 ± 112 ng m$^{-3}$) in Malacca,

30.3–626 ng m$^{-3}$ (233 ± 143 ng m$^{-3}$) in the SCS and 179–454 ng m$^{-3}$ (303 ± 140 ng m$^{-3}$) in SLDP. The dominance

of $C_2$ followed by $C_3$ and $C_4$ diacids is consistent with those in marine aerosols collected from the Sea of Japan

(Mochida et al., 2003a) and western North Pacific (Bikkina et al., 2015).

Azelaic acid ($C_9$) was the most abundant diacid in the range of $C_7$–$C_{11}$ diacids, which is similar to that in marine

aerosols from the southern and western Pacific Ocean (Wang et al., 2006). Such high abundance of $C_9$ diacid

indicates that marine biogenic emissions and subsequent photochemical formation of organic aerosols were likely

significant over the SCS and EIO, because $C_9$ diacid is a photochemical oxidation product of unsaturated fatty acids

emitted from the productive marine regions (Bikkina et al., 2014;Hoque et al., 2017). In addition, the photochemical

oxidation of $C_9$ diacid leads to the generation of its lower homologues including $C_4$–$C_6$ diacids (Kawamura et al.,

1996a;Yang et al., 2008). Thus, the similar abundance of diacids ($C_2$ to $C_6$ and $C_9$) in the SCS and Malacca (Fig. 4)

suggest that the marine biogenic emission was a major source, although the influence of the continental air masses





transported from East and Southeast Asia was also significant. However, the concentrations of diacids over Malacca

were relatively low, which indicated that the influence of polluted air masses transported from Southeast Asia was

lower than those from East Asia.

In general, the abundance of diacids in the EIO-SL and EIO-WI regions was lower than that in the SCS and

Malacca. It was likely because the air masses arrived over EIO-SL and EIO-WI regions mostly originated from the

remote oceanic regions and thus, the marine biogenic emission should be the major potential source of organic

aerosols with no significant influence from the Asian outflow. Interestingly, the concentrations of $C_9$ diacid (average

$68.9 \pm 65.1$ ng m$^{-3}$) in SLDP samples were several times higher than that observed in other regions over the SCS

and EIO. Phthalic acid (Ph acid), a tracer for anthropogenic organic aerosols, was also more abundant in SLDP by

several folds than other regions (Table 1). As noted above, $C_9$ diacid is mostly derived from unsaturated fatty acids

of biogenic origin and has been considered as a marker for biogenic organic aerosols (Kawamura and Gagosian,

1987). Therefore, the high concentration of $C_9$ diacid in SLDP is likely derived by the oxidation of unsaturated

fatty acids from mainland biota. In fact, the terrestrial air masses that enriched with higher plant emissions and

SOA produced upon subsequent oxidation of BVOCs during daytime should be transported onshore during

nighttime by land breeze. In addition, the photochemical breakdown of $C_9$ diacid should be insignificant during

nighttime. Of course, since the enhancement is Ph acid loading was high, we do not preclude the influence of

anthropogenic emissions, particularly fossil fuel combustion, associated with the terrestrial air masses arrived in

SLDP.

Figure 4 shows the temporal variations in selected diacids over the SCS and EIO during the sampling periods.

The temporal patterns of $C_6$ diacid and $C_9$ diacid were similar (Fig. 4). $C_6$ diacid could be produced via the oxidation

of cyclic olefins emitted from anthropogenic sources (e.g., fossil-fuel combustion) (Kawamura et al., 1996b), and

could also be derived from photochemical breakdown of biogenic $C_9$ diacid (Kawamura et al., 1996c). Bikkina et

al. (2015) concluded that the high mass concentrations of $C_9$ in marine aerosols are consistent with the high

biological activity at the ocean surface. Such similar temporal variations in $C_6$ and $C_9$ diacids together with the

results of backward trajectories (Fig. 1b) indicate that the formation of $C_6$ diacid is mostly from the photochemical

breakdown of $C_9$ diacid which should have been derived from marine biogenic emissions.

Based on ultraviolet (UV) irradiation of oleic acid and ozone system, Matsunaga et al. (1999) found a series of

$C_2$ to $C_9$ diacids, with high abundance of $C_9$ diacid. Temporal variations of $C_9$, $C_6$, and $C_2−C_4$ diacids were very

similar (Fig. 4). Such similarities suggest that that photochemical breakdown of $C_9$ might be the major formation

pathway of short-chain diacids such as $C_6$, $C_5$, and $C_4$ diacids. Further aging processes eventually led to the



formation of $C_2$ diacid in the EIO. Thus, the molecular distributions and temporal variations of diacids inferred that

the organic aerosols collected over the EIO should be mainly derived from marine biogenic emissions. However,

the contribution of diacids to WSOC was reduced in the EIO-WI and EIO-SL due to low biological activity in those

oceanic regions as reflected by low $C_9$ diacid loading.

Concentrations of Ph acid over the SCS to EIO were much lower than that in urban Tokyo in summer (average

29 ng m$^{-3}$) (Kawamura and Yasui, 2005) and Chinese cities (90 ng m$^{-3}$) (Ho et al., 2007). Ph acid can be released

directly from fossil fuel combustion (Kawamura and Kaplan, 1987). Secondary formation through atmospheric

oxidation of aromatic hydrocarbons such as naphthalene is also important (Fu et al., 2009;Ho et al., 2010). Such

low levels of Ph acid suggested that the marine aerosols of EIO were not so seriously impacted by continental air

masses from East and Southeast Asia.

### 3.1.2 Oxocarboxylic acids

Oxocarboxylic acids or oxoacids, the intermediate products of the oxidation of monocarboxylic acids, can further

be oxidized to form diacids (Warneck, 2003;Carlton et al., 2007). The concentrations of total oxoacids ranged from

1.48 to 13.2 ng m$^{-3}$ (average $6.51 \pm 3.99$ ng m$^{-3}$) in the SCS, 0.16-2.01 ng m$^{-3}$ ($0.96 \pm 0.46$ ng m$^{-3}$) in the EIO-WI,

0.23–5.06 ng m$^{-3}$ ($1.85 \pm 1.51$ ng m$^{-3}$) in the EIO-SL, 0.29–10.4 ng m$^{-3}$ ($4.22 \pm 3.75$ ng m$^{-3}$) in Malacca and 7.46–

40.3 ng m$^{-3}$ ($24.0 \pm 16.4$ ng m$^{-3}$) in SLDP. The concentrations of total oxoacids are lower to those from Gosan, Jeju

Island, South Korea (average 53 ng m$^{-3}$) (Kawamura et al., 2004) and urban sites in China (45 ng m$^{-3}$) (Ho et al.,

2007). Oxoacids showed a predominance of $\omega C_2$ or $\omega C_3$ in five sampling areas (Fig. 3). The concentration of

oxoacids was the lowest in the EIO-WI compared that in other regions (Table 1). The spatial distributions of $\omega$-

oxoacids and $\alpha$-dicarbonyls showed a pattern: higher at SLDP > SCS > Malacca > EIO-SL > EIO-WI, which was

consistent with those of major diacids ($C_2$, $C_3$, and $C_4$).

Glyoxylic acid ($\omega C_2$) was the most abundant oxoacid followed by pyruvic acid (Pyr) acid (Table 1). All of them

are important intermediates in photo-oxidation processes and are used in the production of low carbon-number

diacids such as $C_2$, $C_3$ and $C_4$ diacids (Hatakeyama et al., 1987). Several studies reported that Pyr is produced by

in-cloud oxidation of isoprene in the atmosphere (Carlton et al., 2006;Carlton et al., 2009) that can be transported

from mainland regions and/or emitted from the ocean surface (Shaw et al., 2010).

### 3.1.3 $\alpha$-Dicarbonyls


The concentrations of total $\alpha$-dicarbonyls varied over a wide range (0.56–20.2 ng m$^{-3}$) with relatively high

abundance in the SCS (0.56–20.2 ng m$^{-3}$, average $5.25 \pm 4.46$ ng m$^{-3}$) and SLDP (5.04–11.5 ng m$^{-3}$, average



7.35±3.58 ng m$^{-3}$). The concentration of total α-dicarbonyls during the cruise was higher than the concentrations of total α-dicarbonyls in super-micron and submicron aerosols collected over the North Pacific in summer (0.51±0.22 ng m$^{-3}$ in more biologically influenced aerosols and 0.66±0.20 ng m$^{-3}$ in less biologically influenced aerosols) (Miyazaki et al., 2010). The abundances of α-dicarbonyls were higher in the SCS and Malacca than those in the EIO-SL and EIO-WI, which are similar to diacids and oxoacids (Fig. 3 and Table 1). The concentrations of both glyoxal (Gly) and methylglyoxal (MeGly) were higher in the SCS and their spatial distributions were consistent with those of diacids. Gly and MeGly can be formed via atmospheric oxidation of isoprene, emitted from the terrestrial vegetation and/or marine phytoplankton (Sorooshian et al., 2009). Besides, Gly is also a photochemical oxidation product of aromatic hydrocarbons from combustion of fossil fuels (Carlton et al., 2007;Volkamer et al., 2007). Both species can be existed in the aerosol phase and further oxidized to form less-volatile organic acids such as Pyr, ω$C_2$, and $C_2$ (Sorooshian et al., 2006). Therefore, the higher abundances of Gly and MeGly over the SCS than in other three regions could be due to enhanced continental outflow as well as the high biological activity in the sea surface.

### 3.2 Relative abundances of diacids and related compounds

The concentrations of diacids and related compounds and their relative abundances in the atmosphere are controlled by the emission of their precursors and subsequent oxidation processes. Pie diagrams of the percentage contributions of individual straight chain diacids to total aliphatic diacids in the different regions over the SCS to EIO (SCS, EIO-WI, EIO-SL, Malacca and SLDP) are depicted in Fig 5. The percentage contributions of LMW-diacids ($C_2$–$C_4$) in total mass concentrations ($\Sigma(C_2$–$C_{12})$) varied from 96.4 to 98.9% (97.6±1.25%) in the SCS, 85.2 to 96.0% (90.6±5.38%) in the EIO-WI, 90.6 to 99.4% (95.0±4.44%) in the EIO-SL, 94.2 to 98.0% (96.1±1.88%) in Malacca, and 72.0 to 95.0% (83.5±11.5%) in SLDP. Interestingly, the relative abundances of $C_2$ to total mass concentrations of $C_2$ to $C_{10}$ diacids ($\Sigma C_2$–$C_{10}$) were similar in four regions (Fig. 5). A significant difference was found in the relative abundance of $C_2$ diacid between SLDP and other four regions with the lower value in SLDP, whereas that of $C_9$ diacid was the opposite.

Figure 6 presents the relative abundances of the sums of short-chain diacids ($C_2$–$C_4$), long-chain diacids ($C_5$–$C_{12}$), unsaturated diacids (M, F, mM, Ph, iPh, and tPh), diacids containing additional functional group (h$C_4$, k$C_3$, and k$C_7$), oxoacids (ω$C_2$−ω$C_9$ and Pyr) and α-dicarbonyls (MeGly, Gly) in the measured total diacids and related compounds in different regions from the SCS to EIO. The relative abundances of short-chain diacids were much higher accounting for about 90% in the SCS and about 60% in the EIO-WI. Long-chain diacids were found be second highest in SLDP and Malacca whereas that of α-dicarbonyls in the EIO-WI and EIO-SL, with high



abundance in the SLDP and EIO-SL, respectively (Fig. 6). Furthermore, the abundance of long-chain diacids was

relatively high in the EIO-WI and EIO-SL but was the lowest in the SCS. The relative abundances of unsaturated

diacids were the highest in the EIO-WI followed by SLDP; while oxoacids were the highest in SLDP, followed by

the EIO-WI, with the lowest values in the SCS and Malacca (Fig. 6). The high abundances of short-chain diacids

indicate that the aerosols over the SCS to EIO were significantly aged during the long-distance transport, while the

high abundances of long-chain diacids and α-dicarbonyls suggest that the contributions from biogenic emissions

were much higher over the EIO and at SLDP. The high atmospheric levels of unsaturated fatty acids and oxoacids

in SLDP infer that the contributions from anthropogenic sources are also significant at SLDP.

### 3.3 Implications for origins and formation pathways

### 3.3.1 Diagnostic mass ratios

Different emission sources and the extent of aging lead to large differences in the concentrations of diacids and

related compounds, and hence, mass concentration ratios of selected species can serve as effective markers for the

identification of their origins and formation/transformation processes. Temporal variations in mass ratios of several

organic acids as well as the relative abundance of $C_2$ to total aliphatic diacids are depicted in Fig. 7. The temporal

trends of $C_3/C_4$, $C_2/C_4$ and $C_2/(\Sigma C_2–C_{12})$ ratios are almost similar. It has been reported that $C_3$ diacid can be produced

from $C_4$ diacid by oxidative reaction; the $C_3/C_4$ ratio can be used as an effective marker to assess the extent of

organic aerosol aging (Kawamura and Ikushima, 1993). The $C_3/C_4$ ratios were reported to be low (0.25–0.44) in

vehicular emissions, because the $C_3$ diacid is thermally unstable to be easily degraded during combustion

(Kawamura and Kaplan, 1987). In contrast, the $C_3/C_4$ ratios reported to be high (range, 1.0–11, average 3.9) in

marine aerosols collected from the North Pacific (including tropics), which have been considered to be

photochemically aged during the atmospheric transport (Kawamura and Sakaguchi, 1999). The $C_3/C_4$ ratios in this

study ranged from 0.3 to 4.6 with an average value of 2.0. They are higher than those in urban aerosols from

Chinese megacities (0.6–1.1, average 0.74) (Ho et al., 2007), Tokyo (0.56–2.9, average 1.6) (Kawamura and

Ikushima, 1993) and New Delhi, India (0.40–1.1; average 0.66 in daytime and 0.58 in nighttime) (Miyazaki et al.,

2009). It is worth noting that both malonic and succinic acids show a net loss from coastal areas to the open oceans,

but the net loss is less for $C_3$ diacid—potentially owing to photochemical processing (Fu et al., 2013). Therefore,

such higher $C_3/C_4$ ratios suggest that the organic aerosols over the SCS to EIO are significantly subjected for

intensive photochemical aging during the atmospheric long-range transport of the outflows from East and Southeast

Asia to remote oceanic regions (Fig. 1). Furthermore, the $C_3/C_4$ ratios increased with the decrease of latitude over





the SCS, suggesting that the secondary formation was enhanced at lower latitudes due to increase temperature and the solar radiation. However, such a trend did not appear during the return voyage, probably due to significant influence of the outflows from mainland China.

The $C_2/C_4$ mass ratios can also be viewed as a marker ratio to assess the extent of organic aerosols aging (Sorooshian et al., 2007). In the SCS, the $C_2/C_4$ ratios (7.6–28.9) showed an increasing trend with the decreasing latitude. The $C_2/C_4$ ratios in the EIO are higher than those reported in Chinese cities (average 7.1) (Ho et al., 2007), Sapporo (average 3.1) (Aggarwal and Kawamura, 2008), and South Korea (average 8.6) (Kundu et al., 2010b). Such a result indicates that $C_2$ could be largely generated by the photochemical degradation of $C_4$ diacid. A

concurrent trend (Fig. 7c) was also obtained for the increased contribution of $C_2$ to total aliphatic diacids ($C_2/\Sigma(C_2$–$C_{12}$)), suggesting more photochemical aging of organic aerosols in low latitudes with stronger solar radiation. Furthermore, in the EIO-WI, some samples near the middle position exhibit low values due to sea source emissions, and the remaining aerosols show relatively high values, which might be affected by both terrestrial and marine emissions.

The $C_2/\Sigma(C_2$–$C_{12}$) ratio has also been considered as an indicator to the degree of organic aerosols aging during long-distance atmospheric transport (Wang et al., 2006). In general, the higher the aerosol aging, the higher the $C_2/\Sigma(C_2$–$C_{12}$) ratio (Kawamura and Sakaguchi, 1999). The $C_2/\Sigma(C_2$–$C_{12}$) was higher (0.76±0.06) in the SCS than that in the EIO-SL (0.74±0.08) and Malacca (0.71±0.11), followed by the EIO-WI (0.65±0.11) and SLDP (0.60±0.11). It is lower than that (0.8±0.04) reported in wintertime and higher than in summertime (0.5±0.01)

Himalayan aerosols (Hegde and Kawamura, 2012). These results indicate that the organic aerosols in the SCS should have been more aged and/or influenced by anthropogenic emissions from East Asia, whereas in the EIO-WI and Malacca the aging of organic aerosols might be less intensive and influenced by only the marine biogenic emissions. Whilst in SLDP, the low $C_2/\Sigma(C_2$–$C_{12}$) should have been driven by the local coastal biota and the terrestrial biogenic and anthropogenic emissions as well as the less aging, particularly in the nighttime.

$C_6$ and Ph acids are mostly derived from cyclic olefins (e.g., cyclohexene) and aromatic hydrocarbons (e.g., naphthalene), respectively, of anthropogenic emissions (Kalberer et al., 2000;Schauer et al., 2001;Ho et al., 2006). In contrast, $C_9$ diacid is a main oxidation product of biogenic unsaturated fatty acids (Rogge et al., 1991;Kawamura and Ikushima, 1993). Hence, the $C_6/C_9$ and Ph/$C_9$ ratios may show special insights into the relative strength of biogenic and anthropogenic sources in the given area. $C_6/C_9$ ratios (Fig. 7f) showed lower values in the EIO (except

sample No. 2), which can be attributed for large influence of marine air masses containing fatty acids like oleic acid, a precursor of $C_9$ diacid (Kawamura and Gagosian, 1987). Temporal variations in Ph/$C_9$ ratios over the SCS



to EIO are shown in Fig. 7g, with the averages of 2.02 in the SCS, 1.56 in the EIO-WI, 1.22 in the EIO-SL, and

0.90 in Malacca. They were lower than that (5.71) reported in wintertime Gosan aerosols from South Korea (Kundu

et al., 2010a), but higher than these in Xi'an summertime (1.7) and wintertime (1.78) aerosols, China (Wang et al.,

2012) and that in the western Pacific (1.41) (Sempéré and Kawamura, 2003). Such comparisons suggest significant

secondary formation of organic aerosols from anthropogenic precursors (e.g., naphthalene) transported from East

and Southeast Asia (Fig. 1). However, $C_6$ and $C_9$ diacids showed a significant correlation (R = 0.51, p < 0.0001)

among total samples, indicating that the former acid should be mainly formed from photo-oxidation of biogenic

unsaturated fatty acids. The higher $Ph/C_9$ ratios found over the SCS than in other regions (Fig. 7g) again suggest

that the contribution from fossil-fuel combustion is also significant, associated with the continental outflow mainly

over the SCS and slightly over the EIO.

Isoprene as well as ethene and ethyne can be significantly emitted into the atmosphere from marine

phytoplankton and other BVOCs like dimethylsulfide (Shaw et al., 2010), and the former significantly produces

MeGly, Gly, and Pyr by oxidation (Carlton et al., 2009). Isoprene is also a dominant VOC emitted from terrestrial

vegetation (Sorooshian et al., 2009). In addition, MeGly, Gly and Pyr can be derived from VOCs emitted from

fossil fuel combustion (Kawamura et al., 1996b). Based on chemical composition data, it is difficult to attribute the

effect of gaseous or heterogeneous reactions occurring in marine atmospheric boundary layer (MABL). However,

on the basis of laboratory experiments, atmospheric oxidation of gaseous isoprene results into the formation of

semi volatile α-dicarbonyls, like Gly and MeGly (Carlton et al., 2007). They further partition into cloud droplets

and/or aqueous aerosols due to their water solubility and involve in aqueous oxidation reactions resulting in $\omega C_2$

and Pyr, which are ultimately converted to $C_2$ diacid (Carlton et al., 2007;Ervens et al., 2004). Hence, the mass

ratios such as $Pyr/C_2$, $\omega C_2/C_2$, $Gly/C_2$, and $MeGly/C_2$, can be used to explore the $C_2$ formation pathway and/or its

origin. In this study, all the ratios were found to peak in the EIO-WI and/or EIO-SL regions (Fig. 7), where the

clean oceanic air masses arrived at the sampling points (Fig. 1). Whereas in the SCS, Malacca and SLDP, those

ratios, except for $\omega C_2/C_2$, were insignificant (Fig. 7). Such spatial distributions of $Pyr/C_2$, $\omega C_2/C_2$, $Gly/C_2$,

$MeGly/C_2$ mass ratios indicate that the organic aerosols loading in the EIO-WI and EIO-SL regions should have

been influenced by the marine biogenic emissions and their subsequent oxidation processes.

Maleic acid (M, *cis*-form) is transformed to fumaric acid (F, *trans*-form) with enhanced photochemical activity

and hence, and the mass ratio of M/F has been considered as a proxy to assess the degree of aerosol aging

(Kawamura and Sakaguchi, 1999). In general, low M/F values reflect secondary oxidation as an important source.

M/F ratios (0.07–0.73) were found to be lower in the EIO-WI than other (0.9–2.3) regions (Table 2). The mean



value of M/F was 1.33 in samples collected in the SCS, 0.29 in the EIO-WI, 1.30 in the EIO-SL, 2.08 in Malacca, and 1.77 in SLDP. They were all lower than those in Xi'an aerosols (summer: 2.22, winter: 2.38), and were about the same or less than those reported at the Korean Gosan site (spring: 1.38, summer: 0.76, autumn: 1.62, winter:

2.21), which have been considered to be aged, indicating that the aging of organic aerosols was more intensive in the EIO-WI, whereas in Malacca and SLDP, the organic aerosols were less significantly aged with significant influenced by local emissions and *in-situ* photochemical oxidation, and thus the *cis-trans* transformation is not significant.

### 3.3.2 Linear correlations

In order to further confirm the origins and formation pathways of diacids and related compounds, we examined the correlations between selected species and also conducted linear regression analyses for mass ratios of the selected chemical specie. Robust relationships were obtained among $C_2$, $C_3$, and $C_4$ diacids in the SCS, EIO-WI and EIO-SL regions (Table 3), suggesting that they should have a common source. A significant correlation was found between relative abundance of $C_2$ ($C_2\%$) and $C_2/C_4$ mass ratio (Fig. 8c), suggesting that the formation of $C_2$ diacid

from photochemical breakdown of $C_4$ diacid was significant. Meanwhile, moderate correlations were also observed for $C_6$ with $C_9$ diacids in the EIO-WI and EIO-SL, and significant correlations over the SCS and Malacca regions. Such positive correlations indicate that the $C_9$ diacid generated by the oxidation of biogenic unsaturated fatty acids should have been subjected to further oxidation during the long-range atmospheric transport resulting in its lower homologues including $C_6$ diacid (Kawamura et al., 1996b;Kawamura et al., 1996c;Kawamura and Sakaguchi,

1999). Pyr, $\omega C_2$ and MeGly showed good correlation in the SCS and Malacca and poor correlations, except for Pyr and MeGly and $\omega C_2$ and Pyr, at the TIO-WI and EIO-SL, respectively, indicating that the organic aerosols derived from BVOCs emitted from the oceanic biota (e.g., isoprene) was more aged in the EIO regions, whereas they might be less aged and should also significantly influenced by anthropogenic emissions and thus, the formation processes of Pyr, $\omega C_2$, and MeGly in the SCS and Malacca regions (Ervens et al., 2008;Bikkina et al., 2014).

Hydroxy succinic acid (malic acid, $hC_4$), $C_4$ and $C_3$ diacids in the SCS correlated well each other ($p < 0.05$; $R^2$ = 0.87 for $hC_4$ *vs* $C_3$ and $R^2$ = 0.78 for $hC_4$ *vs* $C_4$; Fig. 8a and 8b). The formation of $C_3$ diacid from $C_4$ diacid through $hC_4$ as an intermediate has been proposed by (Kawamura and Sakaguchi, 1999) and also observed in a laboratory study (Yang et al., 2008). A very good linear relationship of $hC_4$ diacid with $C_4$ diacid ($R^2$ = 0.85) and $C_2$ diacid ($R^2$ = 0.78) has been reported in forest aerosols from central Europe, which have been attributed to photochemical

production from biogenic unsaturated fatty acids (Kourtchev et al., 2009). Therefore, the good correlations observed among $C_3$, $C_4$ and $hC_4$ in the SCS again support that they were derived from a common source and





significantly aged. A significant linear correlation was also found between methylmalonic (iC$_4$) acid with C$_3$ and

C$_4$ diacids. The branched chain diacids could come from the oxidation of isoprene released from the ocean surface.

Such linear correlations also suggest that the diacids and related compounds were photochemical oxidation

products of isoprene and biogenic unsaturated fatty acids in the MABL. A notable feature observed here is the

significant ($p < 0.05$) correlations between isoprene and/or aromatic hydrocarbon oxidation products (MeGly, Pyr,

and $\omega$C$_2$) and C$_3$ and C$_4$ diacids over Malacca, but not in other regions. Nguyen et al. (2010) found that C$_3$ and C$_4$

diacids were formed from isoprene oxidized by ozone based on the laboratory experiment. Furthermore, it has been

reported that isoprene emission results in significant enhancement of SOA and O$_3$ levels at coastal U.S. sites (Gantt

et al., 2010). Hence, the loading of C$_3$ and C$_4$ diacids also should have been influenced by not only the unsaturated

fatty acids but also BVOCs, at least in Malacca region.

The C$_2$/($\Sigma$C$_2$–C$_{10}$) ratios showed negative relationships with C$_3$ and C$_4$ diacids over the SCS to EIO (Fig. 9a and

9b). This phenomenon again confirms the formation of C$_2$ diacid from C$_3$ and C$_4$ diacids by their photochemical

transformations during long-range atmospheric transport. It is also worth noting that the C$_4$/($\Sigma$C$_2$–C$_{10}$) ratios

showed a linear relationship with glutaric (C$_5$) acid in all regions, except the EIO-WI (Fig. 9c). Additionally, a

similar variation trend is observed for the concentrations of C$_5$ and C$_9$ diacids in all regions, although they were

not significant (C$_9$; Fig. 9d). It is obvious because C$_9$ diacid leads to the formation of all its lower homologues

including C$_4$-C$_6$ diacids as detailed earlier. Thus, the observed linear correlations among the relative abundances of

C$_2$, C$_3$, C$_4$, C$_5$, and C$_9$ infer that the aging of organic aerosols was significant during the study period over the SCS

to EIO.

Finally, to better understand the enhanced emission of unsaturated fatty acids and BVOCs from the biologically

active ocean surface over the SCS to EIO, we presented the satellite image of Chlorophyll-a loading in the EIO in

Fig. 1a. The concentrations of Chlorophyll-a in the SCS and Malacca were high. More interestingly, when high

concentrations of diacids and related compounds were detected (Fig. 4), consistent with high concentrations of

Chlorophyll-a (Fig. 1a). This consistency indicates that high abundances of diacids in the SCS and Malacca

attribute to firstly the oceanic emissions of BVOCs from high biological area and secondly the oxidation during

atmospheric transport, rather than the influence of anthropogenic emissions associated with the Asian outflow.

### 3.4 Fractions of diacids and related compounds in WSOC and OC

The concentrations of water-soluble organic carbon (WSOC) were higher (range 1.12–7.44 ng m$^{-3}$, average 4.43 $\pm$

3.17 ng m$^{-3}$) in the SLDP followed by Malacca (0.09–2.72 ng m$^{-3}$, 1.15$\pm$0.92 ng m$^{-3}$), SCS (0.22–2.31 ng m$^{-3}$, 1.05

$\pm$ 0.65 ng m$^{-3}$), EIO-SL (0.06–1.00 ng m$^{-3}$, 0.35 $\pm$ 0.25 ng m$^{-3}$) and EIO-WI (0.05–0.58 ng m$^{-3}$, 0.15$\pm$0.12 ng m$^{-}$





[3]). The WSOC concentrations over the SCS to EIO are much lower than those in urban aerosols from Tokyo (average 13 µg m$^{-3}$) (Sempére and Kawamura, 1994) and in the biomass burning aerosols from Amazonia (18–51 µg m$^{-3}$) (Kundu et al., 2010a). However, WSOC concentrations in the EIO-SL are comparable to those reported in

marine aerosols from the central Pacific (average 0.30 µg m$^{-3}$) (Hoque and Kawamura, 2016), the western Pacific (average 0.33 µg m$^{-3}$) (Sempéré and Kawamura, 2003), and Hawaii (average 0.39 µg m$^{-3}$) (Hoffman and Duce, 2013). Kawamura et al. (2010) determined lower WSOC concentration (0.04–0.30 µg m$^{-3}$, average 0.18 µg m$^{-3}$) in the Arctic aerosols, which are comparable to those observed in the EIO-WI.

Spatial/temporal distributions of WSOC showed much higher values in the SCS and Malacca than those in the

EIO-WI and EIO-SL regions (Fig. 4j). Higher abundances of WSOC in the SCS and Malacca may be caused by enhanced aging of organic compounds from the surface ocean by high biological activities and then subsequent oxidation to produce more water-soluble organics. In this study, the spatial/temporal trends of diacids and related compounds are generally consistent with that of OC, EC and WSOC, which are high in the SCS and low in other regions. The fractions of some major diacids, oxoacids and α-dicarbonyls in WSOC and OC are presented in Table

5. The total diacids account for 12.0 ± 5.75% (range 2.00–32.4%) of WSOC over the EIO. Whilst, the fractions of oxoacids and α-dicarbonyls in WSOC ranged from 0.03 to 1.22% (average 0.34 ± 0.20%) and 0.02-3.12% (0.60 ± 0.68%), respectively. In addition, a strong linear correlation was found between WSOC and C$_2$ diacid (Table 4), which suggests that the WSOC was mainly generated by secondary formation, rather than biomass-burning emissions associated with the Asian outflow.


## 4 Conclusions

The dicarboxylic acids, oxocarboxylic acids and α-dicarbonyls in marine aerosols collected over the SCS to EIO from 10 March to 26 April 2015 provide better insight on their origins and formation processes. The spatial distribution of diacids generally showed high concentrations in nearshore regions (the SCS, Malacca, SLDP) and

relatively low in the remote regions (the EIO-WI and EIO-SL). Their molecular distributions are characterized by the dominance of C$_2$ diacid. The anthropogenic pollutants under the Asian outflow strongly influenced the concentrations of dicarboxylic acids in the SCS and Malacca whereas the oceanic biogenic emissions and subsequent oxidation influenced the distributions of those species in the EIO-WI and EIO-SL. A close relationship in the temporal variability of C$_9$, C$_6$ and total concentration of short-chain diacids (C$_2$−C$_4$) implies that the formation

of lower diacids via the photochemical breakdown of higher diacids such as C$_9$ diacid, produced by oxidation of

unsaturated fatty acids, was highly significant over the SCS to EIO. The positive correlations obtained between $C_2\%$ and $C_2/C_4$ ratios and negative correlations between $C_2\%$ and $C_3\%$ as well as $C_4\%$ indicate that the formation of $C_2$ from $C_4$ diacid *via* $C_3$ is significant. This finding is also supported by good linear relationships among $C_4$, $hC_4$, and $C_3$ diacids. In addition, the enhanced emissions of biogenic unsaturated fatty acids and VOCs from the

ocean surface and subsequent photochemical oxidation controlled the high and low levels of diacids and related compounds measured in the SCS/Malacca and EIO-WI/EIO-SL, respectively. These results combined with 5-day backward air mass trajectories and the satellite image of Chlorophyll-a suggest the emission of BVOCs from the biologically active ocean surface and their subsequent aging were more significant over the SCS and Malacca than that over the EIO-WI and EIO-SL regions. Finally, SLDP samples were affected by all the sources mentioned

above. In addition, terrestrial biogenic and anthropogenic emissions might have played significant role on the loading of diacids and related compounds at SLDP. Further work on the chemical characterization of the marine organic aerosols at a molecular level by high-solution MS such as Orbitrap and FT-ICR MS are needed to obtain the detailed information on sulfur-containing organics from both biogenic and anthropogenic sources.

***Data availability.*** The dataset for this paper is present in the Tables and Figures and is available upon request from the corresponding author (fupingqing@tju.edu.cn).

***Competing interests***. The authors declare that they have no conflict of interest.

***Author contributions.*** PQF designed this research. Samples were collected by LFW. Laboratory analyses were performed by JY and WYZ. The manuscript was written by JY, PQF and CMP with consultation from all other authors.

***Acknowledgements.*** This work was supported the National Natural Science Foundation of China (Grant Nos. 41625014 and 41805118). We sincerely thank the captain and crews of the R/V "Shiyan 1" during the open research cruise NORC2015-10
supported by NSFC Shiptime Sharing Project.



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





**Table 1.** Summary of concentrations (ng m$^{-3}$) of dicarboxylic acids, oxocarboxylic acids and α-dicarbonyls as well as concentrations (µg m$^{-3}$) of EC and OC in TSP samples collected during the NORC2015-10 cruise over South China Sea to the East Indian Ocean.

| Species | SCS (n = 27) Range | Avg±SD | EIO-WI (n = 20) Range | Avg±SD | EIO-SL (n = 28) Range | Avg±SD | Malacca (n = 9) Range | Avg±SD | SLDP (n = 3) Range | Avg±SD |
|---|---|---|---|---|---|---|---|---|---|---|
| **Dicarboxylic acids** | | | | | | | | | | |
| Oxalic, C$_2$ | 30.3–626 | 233±143 | 4.43–70.1 | 18.2±18.4 | 5.21–232 | 58.2±55.3 | 14.5–373 | 145±112 | 179–454 | 303±140 |
| Malonic, C$_3$ | 2.14–87.9 | 36±23 | 0.34–11.4 | 2.09±2.52 | 0.76–42.1 | 8.19±9.95 | 2.57–74.0 | 24.2±25.4 | 25.9–55.4 | 42.2±15.0 |
| Succinic, C$_4$ | 1.73–50.4 | 18±14 | 0.23–6.4 | 1.35±1.65 | 0.57–20.6 | 4.33±4.86 | 1.18–72.9 | 18.0±22.7 | 14.1–78.5 | 45.2±32.2 |
| Glutaric, C$_5$ | 0.23–8.52 | 2.60±2.15 | bdl–0.79 | 0.23±0.19 | bdl–1.69 | 0.48±0.46 | bdl–9.53 | 3.49±3.70 | 2.94–14.6 | 8.82±5.85 |
| Adipic, C$_6$ | 0.13–4.09 | 1.11±0.93 | 0.17–0.83 | 0.42±0.22 | 0.03–1.18 | 0.31±0.29 | 0.01–3.59 | 1.28±1.36 | 1.95–7.17 | 4.76±2.63 |
| Pimelic, C$_7$ | bdl–1.88 | 0.58±0.60 | bdl–0.22 | 0.03±0.07 | bdl–0.50 | 0.12±0.17 | bdl–1.30 | 0.43±0.46 | bdl–0.45 | 0.28±0.25 |
| Suberic, C$_8$ | bdl | bdl | bdl–0.09 | 0.004±0.0 | bdl–14.3 | 0.51±2.71 | bdl | bdl | bdl | bdl |
| Azelaic, C$_9$ | bdl–6.97 | 1.82±1.57 | bdl–2.22 | 0.45±0.49 | 0.09–15.1 | 1.25±2.75 | 0.32–3.96 | 1.82±1.29 | 11.7–140 | 68.9±65.1 |
| Sebacic, C$_{10}$ | bdl–2.69 | 0.28±0.53 | bdl–0.82 | 0.29±0.24 | bdl–9.06 | 0.57±1.68 | bdl–1.10 | 0.58±0.46 | 0.66–3.43 | 2.00±1.39 |
| Undecanedioic, C$_{11}$ | bdl–0.16 | 0.02±0.05 | bdl–0.69 | 0.16±0.15 | bdl–0.38 | 0.13±0.10 | bdl–0.38 | 0.20±0.15 | bdl–1.64 | 0.62±0.89 |
| Dodecanedioic, C$_{12}$ | bdl | bdl | bdl–0.35 | 0.02±0.08 | bdl | bdl | bdl–0.29 | 0.06±0.11 | bdl | bdl |
| Methylmalonic, iC$_4$ | bdl–2.26 | 0.95±0.65 | bdl–0.29 | 0.01±0.06 | bdl–0.82 | 0.14±0.23 | bdl–3.29 | 0.93±1.06 | 1.02–2.47 | 1.94±0.79 |
| Methylsuccinic, iC$_5$ | 0.30–3.37 | 1.46±0.98 | 0.08–0.54 | 0.35±0.12 | 0.13–1.18 | 0.56±0.30 | 0.21–3.56 | 1.63±1.28 | 2.61–10.1 | 6.99±3.89 |
| Methylglutaric, iC$_6$ | bdl–0.68 | 0.05±0.18 | bdl–0.18 | 0.02±0.05 | bdl | bdl | bdl–0.59 | 0.26±0.25 | 0.67–3.29 | 1.84±1.33 |
| Maleic, M | 0.12–2.70 | 0.93±0.67 | 0.07–0.55 | 0.19±0.11 | 0.08–1.02 | 0.28±0.22 | 0.04–1.42 | 0.77±0.59 | 1.58–4.14 | 2.56±1.38 |
| Fumaric, F | 0.17–2.16 | 0.74±0.48 | 0.16–1.45 | 0.77±0.32 | 0.05–0.82 | 0.28±0.22 | 0.08–1.07 | 0.39±0.33 | 0.81–2.49 | 1.49±0.89 |
| Methylmaleic, mM | 0.17–2.13 | 0.82±0.48 | 0.17–0.76 | 0.39±0.15 | 0.06–0.96 | 0.32±0.21 | 0.11–1.33 | 0.73±0.47 | 2.26–5.49 | 3.80±1.62 |
| Phthalic, Ph | 0.16–6.57 | 2.29±1.89 | bdl–1.25 | 0.50±0.34 | 0.06–2.95 | 0.77±0.59 | 0.26–3.56 | 1.64±1.23 | 8.63–13.2 | 10.8±2.31 |
| Isophthalic, iPh | bdl–0.60 | 0.05±0.16 | bdl–0.37 | 0.02±0.08 | bdl–0.22 | 0.02±0.06 | bdl–0.49 | 0.12±0.20 | 1.08–2.73 | 1.99±0.84 |
| Terephthalic, tPh | bdl–2.12 | 0.22±0.57 | bdl–1.25 | 0.13±0.17 | bdl–0.38 | 0.03±0.09 | bdl–0.59 | 0.10±0.21 | bdl–1.76 | 0.59±1.02 |
| Malic, hC$_4$ | bdl–2.25 | 0.62±0.53 | bdl–0.72 | 0.05±0.17 | bdl–0.81 | 0.13±0.23 | bdl–1.05 | 0.31±0.35 | 0.88–1.95 | 1.45±0.54 |
| Ketomalonic, kC$_3$ | bdl–5.84 | 1.76±1.34 | bdl–0.83 | 0.22±0.22 | 0.05–2.10 | 0.54±0.47 | 0.15–2.19 | 1.00±0.90 | 1.15–2.75 | 2.06±0.82 |
| 4-Ketopimelic, kC7 | bdl–7.75 | 1.85±1.70 | bdl–0.63 | 0.12±0.19 | bdl–1.87 | 0.45±0.56 | 0.17–9.27 | 3.59±3.68 | 1.24–3.34 | 2.50±1.10 |
| Subtotal | 44.6–759 | 305±186 | 8.28–96.3 | 26.1±23.3 | 8.55–294 | 77.6±73.1 | 21.9–501 | 207±158 | 259–708 | 514±231 |
| **Oxocarboxylic acids** | | | | | | | | | | |
| Pyruvic, Pyr | 0.01–1.70 | 0.60±0.42 | bdl–0.70 | 0.30±0.17 | 0.11–0.75 | 0.33±0.19 | bdl–1.23 | 0.46±0.44 | 0.83–1.62 | 1.13±0.42 |
| Glyoxylic, ωC$_2$ | 0.41–9.23 | 2.68±1.87 | bdl–0.75 | 0.23±0.23 | 0.09–2.89 | 0.97±0.86 | 0.003–4.92 | 2.05±2.08 | 3.56–8.65 | 6.28±2.56 |
| 3-Oxopropanoic, ωC3 | 0.14–2.53 | 0.86±0.56 | 0.06–0.65 | 0.27±0.14 | bdl–0.75 | 0.25±0.26 | bdl–2.01 | 0.56±0.61 | 0.55–2.16 | 1.13±0.90 |
| 4-Oxobutanoic, ωC4 | bdl–1.09 | 0.29±0.41 | bdl | bdl | bdl–1.42 | 0.06±0.27 | bdl–2.66 | 0.72±1.02 | 1.41–9.27 | 5.10±3.95 |
| 5-Oxopentanoic, ωC5 | bdl–0.09 | 0.01±0.03 | 0.05–0.39 | 0.13±0.08 | bdl–0.15 | 0.04±0.03 | bdl–0.15 | 0.04±0.05 | 0.07–0.21 | 0.16±0.07 |
| 7-Oxoheptanoic, ωC7 | bdl–0.23 | 0.23±0 | bdl | bdl | bdl | bdl | bdl | bdl | bdl | bdl |
| 8-Oxooctanoic, ωC8 | bdl–3.94 | 1.37±1.44 | bdl | bdl | bdl | bdl | bdl–0.23 | 0.07±0.10 | bdl–16.1 | 8.47±8.07 |
| 9-Oxononanoic, ωC9 | bdl–1.89 | 0.69±0.65 | bdl–0.29 | 0.03±0.07 | bdl–1.41 | 0.19±0.36 | bdl–0.88 | 0.32±0.32 | 0.80–2.38 | 1.72±0.83 |
| Subtotal | 1.48–13.2 | 6.51±3.99 | 0.16–2.01 | 0.96±0.46 | 0.23–5.06 | 1.85±1.51 | 0.29–10.4 | 4.22±3.75 | 7.46–40.3 | 24.0±16.4 |
| **α-Dicarbonyls** | | | | | | | | | | |
| Glyoxal, Gly | 0.05–7.70 | 2.28±2.26 | 0.20–3.20 | 1.64±0.67 | 0.11–2.02 | 0.87±0.40 | 0.26–0.87 | 0.52±0.23 | 2.70–7.04 | 4.15±2.50 |
| Methylglyoxal, MeGly | 0.29–20.0 | 2.97±3.72 | 0.88–3.01 | 1.66±0.54 | 0.09–7.45 | 1.22±1.31 | 0.93–4.01 | 1.81±0.93 | 2.31–4.45 | 3.20±1.11 |
| Subtotal | 0.56–20.2 | 5.25±4.46 | 1.08–5.29 | 3.31±1.01 | 0.95–8.86 | 2.08±1.47 | 1.18–4.79 | 2.34±1.06 | 5.04–11.5 | 7.35±3.58 |
| OC | 0.18–2.63 | 1.26±0.67 | 0.02–0.66 | 0.27±0.19 | 0.06–1.69 | 0.06±0.45 | 0.28–4.44 | 2.02±1.54 | 3.57–20.9 | 12.8±8.73 |
| EC | 0.05–0.75 | 0.27±0.15 | 0.01–0.13 | 0.05±0.05 | 0.04–0.91 | 0.04±0.18 | 0.07–0.63 | 0.24±0.18 | 1.67–7.57 | 4.24±3.02 |

bdl: below detection limit, which is ca. 0.005 ng m$^{-3}$ for the target compounds.






**Table 2.** Diagnostic mass ratios of selected diacids and related compounds in TSP samples collected during the NORC2015-10 cruise.

| Mass Ratio | SCS | EIO-WI | EIO-SL | Malacca | SLDP |
|---|---|---|---|---|---|
| | | | Avg ± SD | | |
| $C_2/C_4$ | 16.2±8.14 | 17.1±10.6 | 16.1±6.72 | 15.1±12.4 | 8.28±3.80 |
| $C_3/C_4$ | 2.18±0.95 | 1.79±0.90 | 2.05±0.80 | 1.99±1.14 | 1.23±0.63 |
| $C_2/\omega C_2$ | 101±69.8 | 146±220 | 69.7±41.2 | 721±1678 | 48.1±5.79 |
| $C_2/Pyr$ | 580±499 | 70.3±63.7 | 193±191 | 294±254 | 267±71.0 |
| $C_2/\Sigma(C_2–C_{12})$ | 0.80±0.05 | 0.76±0.07 | 0.78±0.07 | 0.76±0.11 | 0.65±0.12 |
| $C_3/\Sigma(C_2–C_{12})$ | 0.12±0.04 | 0.09±0.02 | 0.11±0.03 | 0.12±0.05 | 0.09±0.02 |
| $C_4/\Sigma(C_2–C_{12})$ | 0.06±0.02 | 0.06±0.03 | 0.06±0.04 | 0.08±0.06 | 0.09±0.03 |
| $C_6/C_9$ | 0.76±0.93 | 1.17±0.76 | 0.49±0.52 | 0.49±0.39 | 0.11±0.07 |
| $Ph/C_9$ | 2.02±4.26 | 1.66±1.33 | 1.22±1.19 | 0.90±0.21 | 0.34±0.34 |
| M/F | 1.33±0.52 | 0.29±0.19 | 1.30±0.59 | 2.08±1.72 | 1.77±0.17 |
| $C_2/MeGly$ | 129±156 | 11.3±10.6 | 57.9±40.8 | 81.9±55.5 | 92.1±13.1 |
| $C_2/Gly$ | 681±1466 | 14.6±15.2 | 127±196 | 295±184 | 77.4±21.2 |
| Gly/MeGly | 0.86±0.67 | 1.02±0.45 | 1.43±2.31 | 0.31±0.12 | 1.24±0.32 |





**Table 3.** Correlation matrices of selected diacids and related compounds in TSP collected during the NORC2015-10 cruise.

| SCS (n=27) | $C_2$ | $C_3$ | $C_4$ | $C_5$ | $C_6$ | $C_9$ | Ph | Pyr | $\omega C_2$ | MeGly |
|---|---|---|---|---|---|---|---|---|---|---|
| $C_2$ | 1.00 ** | | | | | | | | | |
| $C_3$ | 0.85** | 1.00 | | | | | | | | |
| $C_4$ | 0.79 ** | 0.88 ** | 1.00 | | | | | | | |
| $C_5$ | 0.86** | 0.73 ** | 0.87** | 1.00 | | | | | | |
| $C_6$ | 0.87** | 0.67 ** | 0.75 ** | 0.95 ** | 1.00 | | | | | |
| $C_9$ | 0.67** | 0.38 | 0.36 | 0.70 ** | 0.78** | 1.00 | | | | |
| Ph | 0.70 ** | 0.54 ** | 0.72 ** | 0.86 ** | 0.84 ** | 0.59 ** | 1.00 | | | |
| Pyr | 0.50 ** | 0.44 * | 0.55 ** | 0.52 ** | 0.43 * | 0.19 | 0.52 ** | 1.00 | | |
| $\omega C_2$ | 0.88 ** | 0.71 ** | 0.68 ** | 0.80 ** | 0.79 ** | 0.68 ** | 0.69 ** | 0.66** | 1.00 | |
| MeGly | 0.65 ** | 0.40* | 0.30 | 0.42* | 0.49 * | 0.50 ** | 0.50** | 0.59 ** | 0.80** | 1.00 |

| EIO-WI (n=20) | $C_2$ | $C_3$ | $C_4$ | $C_5$ | $C_6$ | $C_9$ | Ph | Pyr | $\omega C_2$ | MeGly |
|---|---|---|---|---|---|---|---|---|---|---|
| $C_2$ | 1.00 | | | | | | | | | |
| $C_3$ | 0.90** | 1.00 | | | | | | | | |
| $C_4$ | 0.89 ** | 0.94 ** | 1.00 | | | | | | | |
| $C_5$ | 0.85 ** | 0.83 ** | 0.76** | 1.00 | | | | | | |
| $C_6$ | 0.26 | 0.37 | 0.41 | 0.41 | 1.00 | | | | | |
| $C_9$ | 0.34 | 0.36 | 0.36 | 0.39 | 0.51 * | 1.00 | | | | |
| Ph | 0.11 | 0.29 | 0.18 | 0.51 * | 0.67** | 0.40 | 1.00 | | | |
| Pyr | 0.10 | 0.13 | 0.14 | 0.17 | 0.04 | 0.33 | 0.04 | 1.00 | | |
| $\omega C_2$ | 0.68 ** | 0.51* | 0.53 * | 0.71** | –0.04 | 0.40 | 0.12 | 0.23 | 1.00 | |
| MeGly | 0.11 | 0.13 | 0.13 | 0.00 | 0.21 | 0.28 | –0.11 | 0.61 ** | –0.12 | 1.00 |

| EIO-SL (n=28) | $C_2$ | $C_3$ | $C_4$ | $C_5$ | $C_6$ | $C_9$ | Ph | Pyr | $\omega C_2$ | MeGly |
|---|---|---|---|---|---|---|---|---|---|---|
| $C_2$ | 1.00 | | | | | | | | | |
| $C_3$ | 0.89** | 1.00 | | | | | | | | |
| $C_4$ | 0.86 ** | 0.94 ** | 1.00 | | | | | | | |
| $C_5$ | 0.73 ** | 0.75** | 0.78** | 1.00 | | | | | | |
| $C_6$ | 0.41 * | 0.31 | 0.49 ** | 0.44 | 1.00 | | | | | |
| $C_9$ | 0.31 | 0.14 | 0.20 | 0.31 | 0.39* | 1.00 | | | | |
| Ph | 0.53 ** | 0.35 | 0.43 * | 0.52 ** | 0.70 ** | 0.79 ** | 1.00 | | | |
| Pyr | 0.40 * | 0.20 | 0.38 * | 0.24 | 0.38 * | 0.22 | 0.27 | 1.00 | | |
| $\omega C_2$ | 0.79 ** | 0.55** | 0.65 ** | 0.61** | 0.50 ** | 0.51** | 0.69** | 0.64** | 1.00 | |
| MeGly | 0.21 | 0.21 | 0.09 | 0.11 | 0.14 | 0.07 | 0.19 | 0.32 | 0.14 | 1.00 |





| Malacca (n=9) | $C_2$ | $C_3$ | $C_4$ | $C_5$ | $C_6$ | $C_9$ | Ph | Pyr | $\omega C_2$ | MeGly |
|---|---|---|---|---|---|---|---|---|---|---|
| $C_2$ | 1.00 | | | | | | | | | |
| $C_3$ | 0.65 | 1.00 | | | | | | | | |
| $C_4$ | 0.45 | 0.88 ** | 1.00 | | | | | | | |
| $C_5$ | 0.69* | 0.76 * | 0.87** | 1.00 | | | | | | |
| $C_6$ | 0.89 ** | 0.64 | 0.56 | 0.85 ** | 1.00 | | | | | |
| $C_9$ | 0.87 ** | 0.71 * | 0.60 | 0.81 ** | 0.96** | 1.00 | | | | |
| Ph | 0.84 ** | 0.87 ** | 0.80 * | 0.89 ** | 0.90 ** | 0.94 ** | 1.00 | | | |
| Pyr | 0.55 | 0.71 * | 0.83 ** | 0.79 * | 0.60 | 0.56 | 0.71 * | 1.00 | | |
| $\omega C_2$ | 0.80 ** | 0.75 * | 0.80 ** | 0.97 ** | 0.91 ** | 0.86** | 0.91 ** | 0.84** | 1.00 | |
| MeGly | 0.36 | 0.79* | 0.94 ** | 0.83** | 0.50 | 0.52 | 0.74 * | 0.67 | 0.71* | 1.00 |

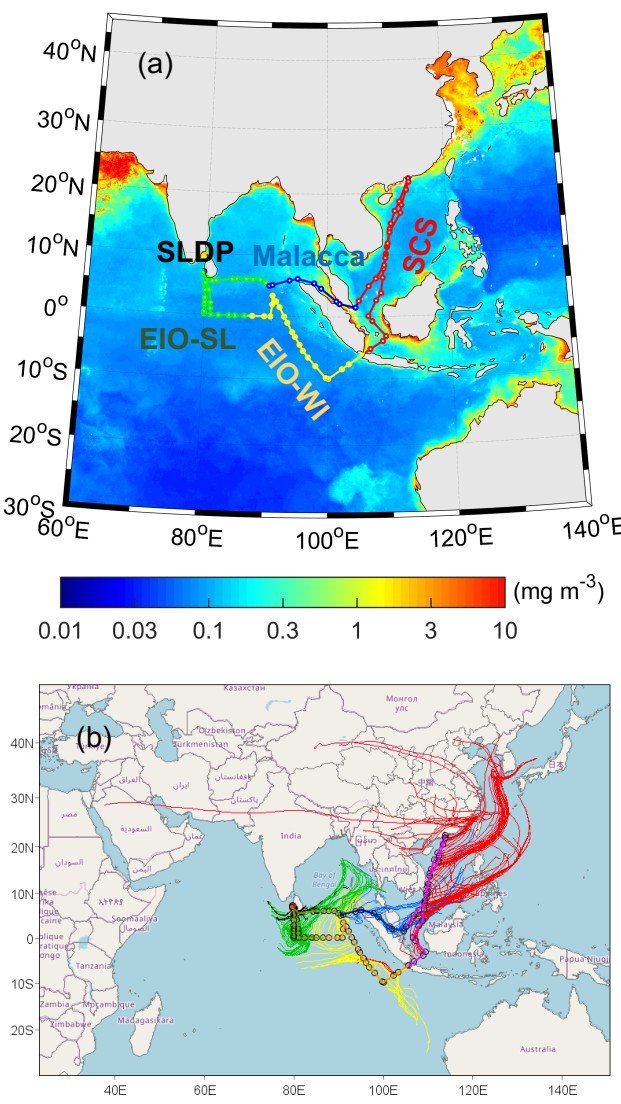

**Figure 1.** (a) Cruise tracks used for aerosols collection in the South China Sea and the East Indian Ocean during 10[th] March to 26[th] April 2015. The area where the red line is located is South China Sea (SCS), the yellow line is East Indian Ocean near western Indonesia (EIO-WI), the green line is East Indian Ocean near Sri Lanka (EIO-SL), the blue line is Malacca, and the red solid origins are the collection positions of three samples of Sri Lanka docking point (SLDP). The base map reflects an average composite of Chla concentrations in March–April 2015 obtained from the NASA website at https://modis.gsfc.nasa.gov/data/dataprod/chlor a.php. (b) Air mass backward trajectories for the sampling days at arrival heights 100 m.





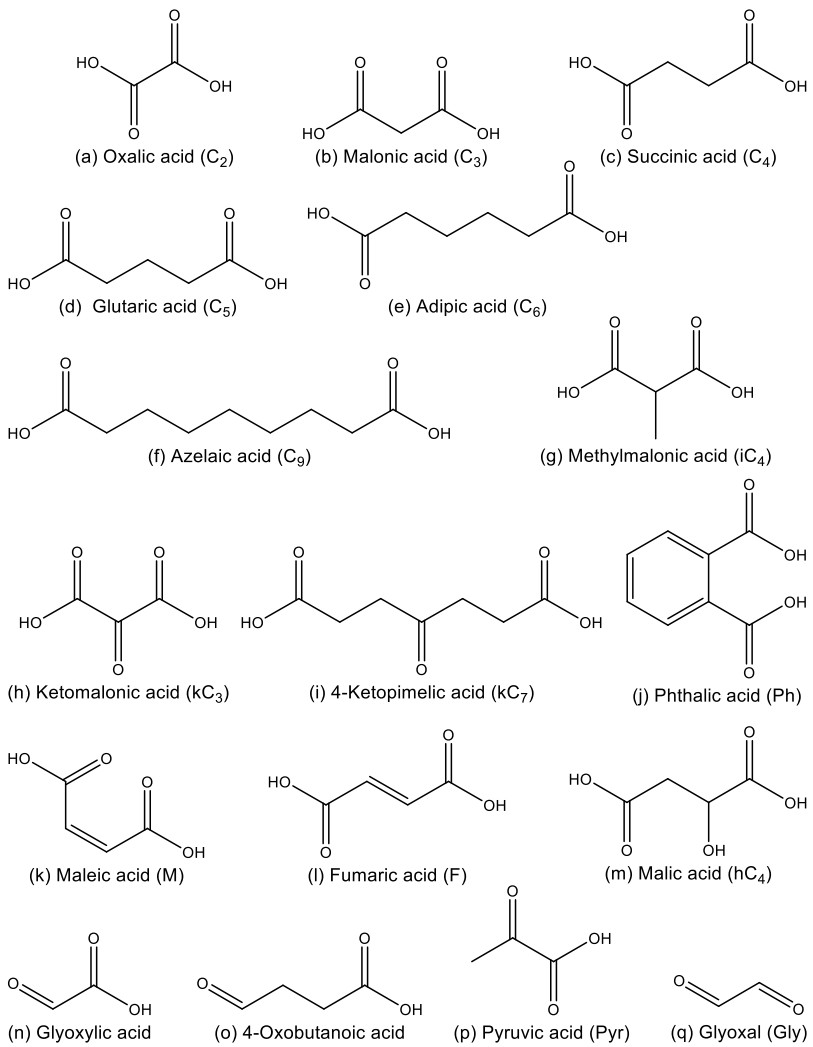

**Figure 2.** Chemical structures of selected dicarboxylic acids and other major organic compounds detected in the marine aerosols.



**Figure 3.** Molecular distributions of dicarboxylic acids and related compounds in TSP samples from (a) SCS, (b) EIO-WI, (c) EIO-SL, (d) Malacca, and (e) SLDP during 10th March to 26th April 2015.






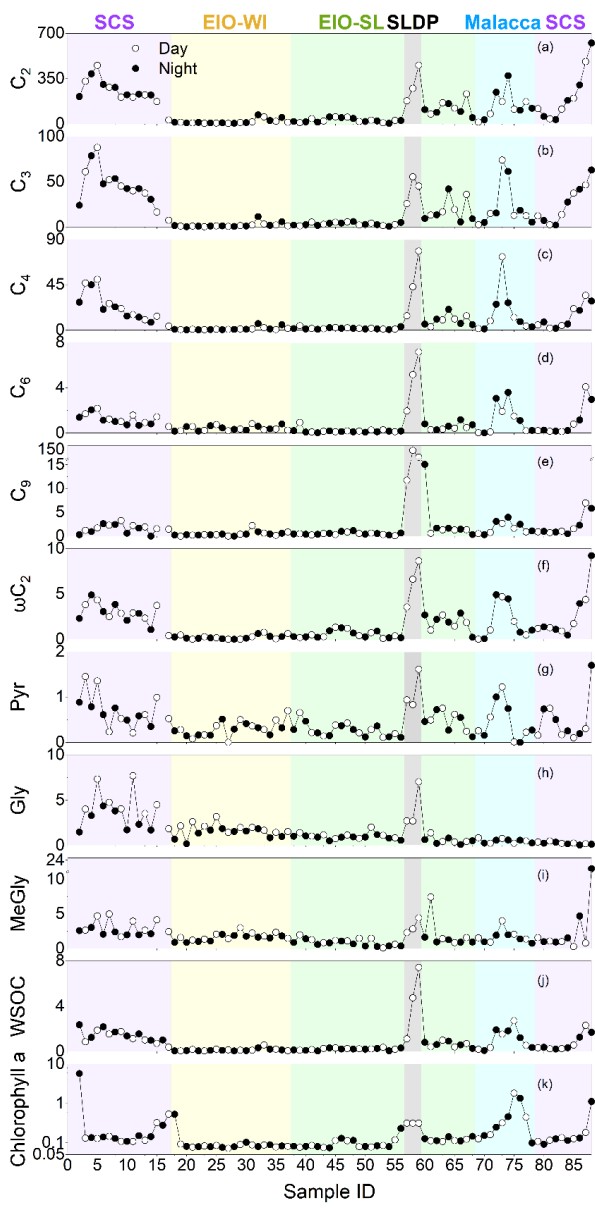


**Figure 4.** Daily variations in the concentrations of selected organic acids and water-soluble organic carbon (WSOC) in the TSP aerosols in South China Sea and the East Indian Ocean from 10th March to 26th April 2015 (all parameters are in ng m$^{-3}$ except WSOC is in μg m$^{-3}$ and Chlorophyll a is in mg m$^{-3}$).

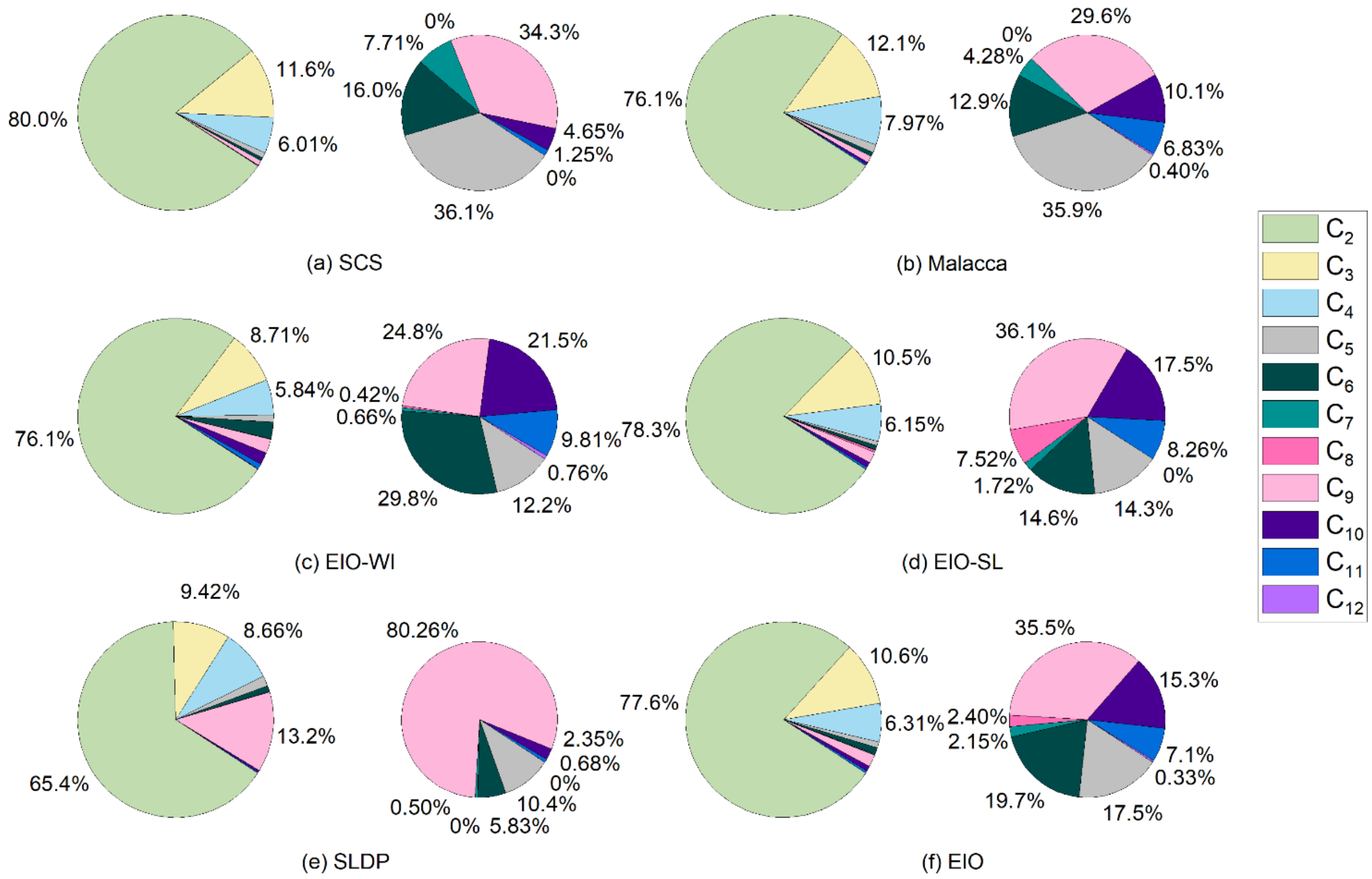


**Figure 5.** Pie charts showing the percentage contribution of individual diacid to total aliphatic homologous diacids ($\Sigma C_2$–$C_{12}$) in different sea areas aerosols collected during the NORC2015-10 cruise.





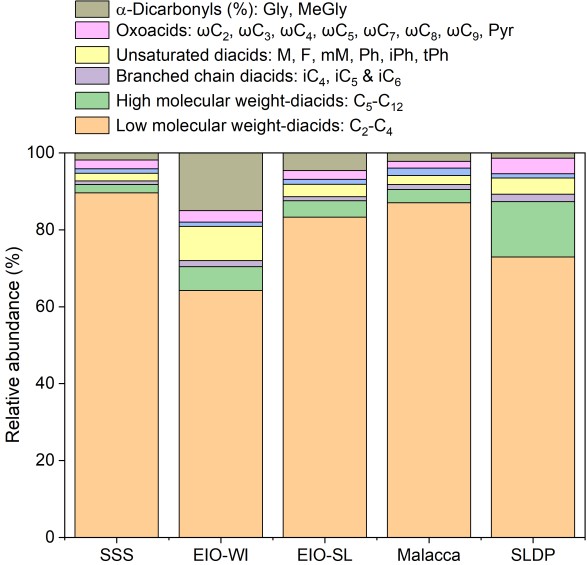


**Figure 6.** Relative abundances of individual compound class in the total diacids, oxoacids, and α-dicarbonyls in different sampling area aerosols collected during the NORC2015-10 cruise.



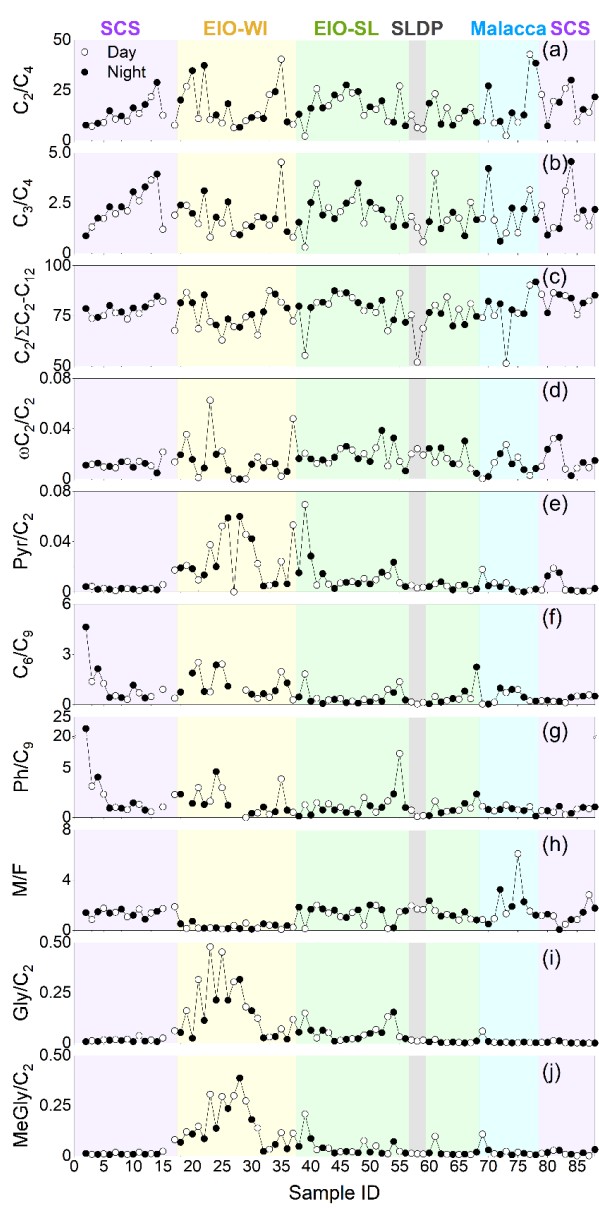


**Figure 7.** Daily variations in the concentration ratios of (a) $C_2/C_4$, (b) $C_3/C_4$, (c) $C_2/\Sigma C_2$–$C_{12}$, (d) $C_2/\omega C_2$, (e) $C_6/C_9$, (f) $C_6/C_9$, (g) Ph/$C_9$, (h) M/F, (i) $C_2$/Gly, and (j) $C_2$/MeGly in TSP samples collected during the NORC2015-10 cruise.






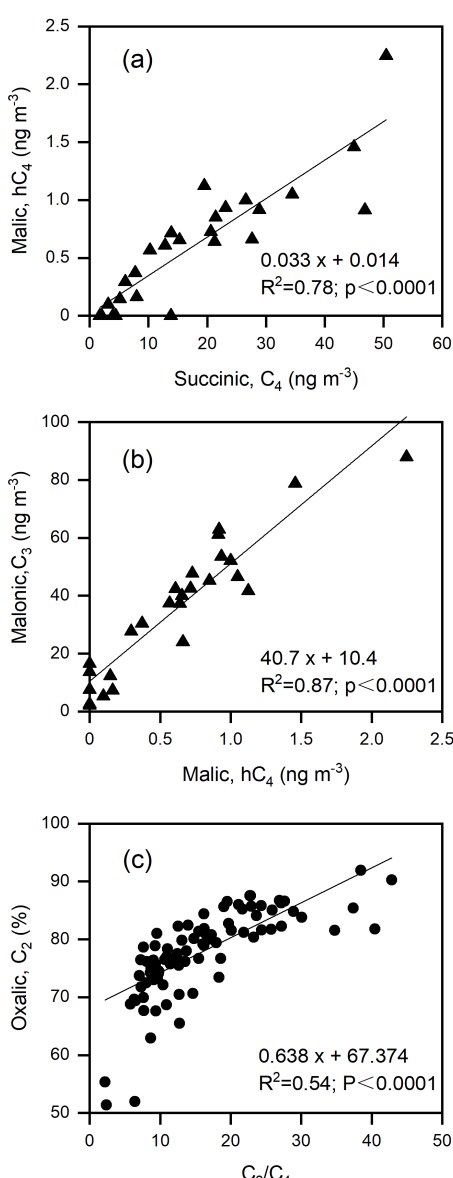

**Figure 8.** Scatterplots for (a) malic versus succinic acids, (b) malonic versus malic acids in aerosols collected over the SCS; (c) relative abundance of oxalic acid in total diacid mass versus $C_2/C_4$ ratios in aerosols collected over the all sea areas.




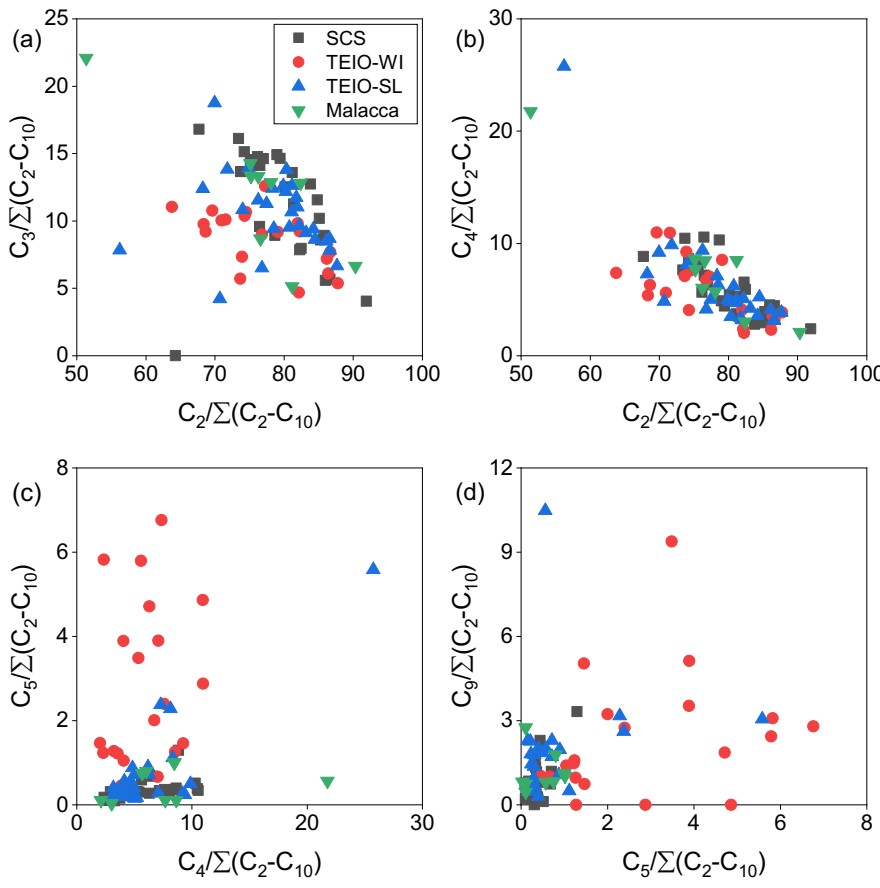


**Figure 9.** Scatterplots between the relative abundances of $C_2$ in their total aliphatic homologues ($\Sigma C_2$–$C_{10}$) with that of (a) malonic acid and (b) succinic acid. Likewise, scatterplots showing the relative abundances of $C_5$ in $\Sigma C_2$–$C_{10}$ with those of (c) $C_4$ and (d) $C_9$ diacids in TSP samples collected during the NORC2015-10 cruise.