# Peer review of "Molecular and spatial distributions of dicarboxylic acids, oxocarboxylic acids and $\alpha$ -dicarbonyls in marine aerosols from the South China Sea to East Indian Ocean"

_Atmospheric Chemistry and Physics, 2019_

## Referee Comment (RC1) · Anonymous Referee #1 · 12 Jan 2020

This observation-base study presents the data of organic aerosol species from a cruise campaign from the South China Sea to the eastern Indian Ocean. It shows the spatial variations of dicarboxylic acids, oxocarboxylic acids and $\alpha$-dicarbonyls in the marine aerosols in the investigated oceanic areas. It also discussed their sources and major influence by the oceanic emissions and long-range transport. It could be accepted for publication in ACP after revision.

1. I would suggest the authors to re-organize the manuscript in order the present their findings in a clearer way. In the current version, it was not easy for me to follow and

to understand the major findings. Some part(s), e.g. their fraction in water-soluble organic carbon, can be moved above to a part maybe mainly describing the overview of data. If possible, the data of ions as well as organic carbon, elemental carbon may be shown there to let the readers quickly get an overview of the data. The authors should also re-organize the discussions for a better presentation of their results and conclusions. For example, the ratios of C3/C4 dicarboxylic acids and their correlations are separated into two different parts, which should be merged. These discussions are highly related.

2. I would suggest the authors to polish their findings. What are the major findings in this observation? What information would they like to bring to the readers?

3. Be very careful to deal with the correlations and ratios, I found they may suggest different/contradictory conclusions in the discussion.

3. The authors should pay attention to their citations. In these oceanic areas, there are some other observations on organic aerosols which should be inspirational for the authors when undergoing their discussions.

4. Some figures can be moved to the supporting information. For example, Figure 2 only shows the chemical structures of the diacids. It is hard to get information from Figure 9 efficiently. Figure 8 could also be moved to the SI.

5. L301-305: I do not understand why? From the ratios, we could say these aerosol particles are aged but it is hard to know if they are influenced by marine biota or continental anthropogenic emissions.

6. L361-365: The sentence is too long and hard to be followed. Please rephrase it.

7. Line 394-396: I do not see the high Chlorophyall-a concentrations in the SCS in the satellite image (Figure 1a). A close look at a special case of some samples (e.g. 55-60) would be necessary.

8. Minor errors: L210: it should be "lower than" L298-299: should the sentence be

"the more the aerosol particles are aged, the higher the ratios are"? L311: "attribute for"should be "attribute to"

---

## Referee Comment (RC2) · Anonymous Referee #2 · 15 Jan 2020

The manuscript studied the distributions of dicarboxylic acids, oxocarboxylic acids and α-dicarbonyls in marine aerosols during a cruise from 10 March to 26 April 2015. The cruise area is over South China Sea to East Indian Ocean. There were many samples collected, and the analysis were based on four regions, SCS, EIO-WI, EIO-SL, and SLDP. Through the different concentrations and ratios of dicarboxylic acids, oxoacids and α-dicarbonyls, their sources and possible formation pathways in each studied region were discussed. The work in this manuscript is very important. However, the article still needs to be major revised and then can be published on Atmos. Chem. Phys.

Major Comments:

1. The main meaningful was not very clear during the part of Introduction. The author should give more discussion about this cruise especially the important of this studied area in the Introduction.

2. There are many analyses and data in the manuscript. It is very difficult for readers to understand the information present in the article because of the illogical.

Part **3.1.1 Dicarboxylic acids**:

There are many discussions about C9 in paragraph 4, 5, 6, 7, and discussions about Ph in paragraph 5 and 8. The whole part of 3.1.1 is very illogical, and I can't catch the important point and main results. The author should analysis the main connection between these dicarboxylic acids and different areas, give more clearly analysis. For example, form the analysis in the manuscript, the most important dicarboxylic acid is the C9 which have relation with C2-C4, C6, and Ph. The author can put these results together and give the discussion, then give the main point of these results.

Part **3.1.2 Oxocarboxylic acids**:

The meaning of oxocarboxylic acids in the secondary paragraph should be discussed firstly in this part.

Part **3.3.1 Diagnostic mass ratios**:

The data of C3/C4, C2/C4, C2/Σ(C2-12), and M/F can give the information of organic aerosols aging. And they can be put together to give the discussion which can be more clearly.

Part **3.3.2 Linear correlations**:

This part can be put in the supporting information, the authors just give the result when other parts need to be supported. For example, line 443-446.

3. The authors just compare their data with references, but the discussions are not enough.

Line 158-159, Line 161-164, Line 220 "Oxoacids showed a predominance of ωC2 or ωC3 in five sampling areas (Fig. 3)." Line 221-224,

4. There are many repetitions in the article not only the example below.

The sources of C9 and the relation of C9 with other carboxylic acids were discussed repeatedly in part 3.1.1.

Part 3.3.1, the meaning of C6, Ph, MeGly, Gly, Pyr, Isoprene has been given in the former part, delete the repetitions.

Minor commets:

Line 254 "C10" to "C12".

Line 386 delete "isoprene and/or aromatic hydrocarbon oxidation products".

Line 287, "It is worth noting that both C3 and C4 acids show a net loss…"

Line 202, delete one "that".

---

## Author Comment (AC1) · 8 Mar 2020

**Responses to Reviewer' comments**

We appreciate the reviewers for their thorough reading and thoughtful comments and suggestions, which greatly improve the quality of the manuscript. We revised the MS accordingly. The point-to-point responses to all the comments are given below in blue.

Reviewer #1 (Formal Review for Author (shown to author)):

This observation-base study presents the data of organic aerosol species from a cruise campaign from the South China Sea to the eastern Indian Ocean. It shows the spatial variations of dicarboxylic acids, oxocarboxylic acids and α-dicarbonyls in the marine aerosols in the investigated oceanic areas. It also discussed their sources and major influence by the oceanic emissions and long-range transport. It could be accepted for publication in ACP after revision.

**Response:** We thank the reviewer for the comments and the recommendation.

Major comments:

1. I would suggest the authors to re-organize the manuscript in order the present their findings in a clearer way. In the current version, it was not easy for me to follow and to understand the major findings. Some part(s), e.g. their fraction in water-soluble organic carbon, can be moved above to a part maybe mainly describing the overview of data. If possible, the data of ions as well as organic carbon, elemental carbon may be shown there to let the readers quickly get an overview of the data. The authors should also re-organize the discussions for a better presentation of their results and conclusions. For example, the ratios of $C_3/C_4$ dicarboxylic acids and their correlations are separated into two different parts, which should be merged. These discussions are highly related.

**Response:** Following the reviewer's suggestion, we have added the description of OC and EC in the revised MS as a sub-section 3.1. Also, the description of WSOC data has been moved to sub-section 3.1 from the 3.5. In addition, the WSOC data has been given in Table 1. The correlations of $C_2$, $C_3$, and $C_4$, and those of $C_2$% and $C_2/C_4$, as well as Pyr, $\omega C_2$, and MeGly, and the relevant description of chlorophyll-a in the original sub-section "3.3.2 Linear correlations" have been combined and placed in original sub-section "3.3.1 Diagnostic mass ratios". The rest of original sub-section "3.3.2 Linear correlations" has been moved to the supporting information.

2. I would suggest the authors to polish their findings. What are the major findings in this observation? What information would they like to bring to the readers?

**Response:** We have improved the discussion of our results to make our findings such as the origins and formation processes (mainly marine BVOCs and *in-situ* secondary formation) over the SCS-EIO region clear to the readers. Please see the abstract and conclusions in the revised MS.

3. Be very careful to deal with the correlations and ratios, I found they may suggest different/contradictory conclusions in the discussion.

**Response:** We have carefully interpreted the mass ratios and correlations between selected species in assessing the origins and secondary formation/transformations of diacids and related compounds and discussed in detail, without leaving any contradictory view. Please see the sub-section 3.4 in the revised MS.

3. The authors should pay attention to their citations. In these oceanic areas, there are some other observations on organic aerosols which should be inspirational for the authors when undergoing their discussions.

**Response:** We have tried our best to make modifications to the original text and have added some discussions in the revision. The revised content is as followed:

"The dominance of $C_2$ followed by $C_3$ and $C_4$ diacids is consistent with those coastal marine aerosols (Kundu et al., 2010;Kunwar and Kawamura, 2014) and remote marine aerosols from the western North Pacific (Bikkina et al., 2015), suggesting similar formation processes of dicarboxylic acids in the atmosphere." (see Page 8, Line 213–216).

We also added the following sentences in the revised manuscript.

"For the aerosol samples of SCS, the 5-day back trajectories (Fig. 1b) showed that the air masses were originated from East Asia, whereas the air masses of Malacca were delivered from Southeast Asia. The concentration of diacids and related compounds were highest in SCS and second highest in Malacca, due to increased anthropogenic activities through long-range atmospheric transport in East Asia and Southeast Asia, respectively." (see Page 8, Line 206–210)

"The spatial distributions of total oxoacids and α-dicarbonyls were similar to those of total diacids, indicating that they were similar in origin or formation mechanism (Kunwar et al., 2017). Based on the backward trajectory analysis, we identified four source regions of diacids and related compounds: (a) East Asia, (b) Southeast Asia, (c) the Bay of Bengal, and (d) the East Indian Ocean. When air masses originated from East Asia and Southeast Asia and their

coastal areas (i.e. SCS and Malacca), the concentrations of these compounds were high; when air masses were mainly derived from the East Indian Ocean (i.e. EIO-WI), their concentrations was the lowest; and when the air masses originated from the Bay of Bengal and the East Indian Ocean (i.e. EIO-SL), the concentration of these compounds was between the former two." (see Page 12, Line 327–335)

4. Some figures can be moved to the supporting information. For example, Figure 2 only shows the chemical structures of the diacids. It is hard to get information from Figure 9 efficiently. Figure 8 could also be moved to the SI.

**Response:** In the revision, we have moved Figure 2, Figure 8 and Figure 9 to the supporting information (SI).

5. L301-305: I do not understand why? From the ratios, we could say these aerosol particles are aged but it is hard to know if they are influenced by marine biota or continental anthropogenic emissions.

**Response:** To avoid such confusion/misunderstanding to the reader, we modified the discussion by removing the following sentence: "Further ---- emissions." and by detailing the interpretations of the ratios and correlations of the selected species. In fact, the air masses transported from the continental regions are aged (confirmed by $C_2$%) and also mainly influenced by the marine biota emissions (confirmed by the mass ratios and correlations between species, e.g., $C_6$ and $C_9$ diacids). We detailed these points in the revised MS (please see sub-section 3.4).

6. L361-365: The sentence is too long and hard to be followed. Please rephrase it.

**Response:** Now the revised sentences are as follows:
"In addition, there were good correlations among Pyr, $\omega C_2$ and MeGly in the SCS and Malacca, while the correlations were poor in the samples from EIO-WI and EIO-SL (except for Pyr and MeGly in EIO-WI, and $\omega C_2$ and Pyr in EIO-SL). The results showed that organic aerosols produced by BVOCs (e.g. isoprene) emitted from the ocean surface were more aged in the EIO-WI and EIO-SL than the SCS and Malacca where were affected by anthropogenic emissions." (see Page 15, Line 437–441)

7. Line 394-396: I do not see the high Chlorophyall-a concentrations in the SCS in the satellite image (Figure 1a). A close look at a special case of some samples (e.g. 55-60) would be necessary.

**Response:** The manuscript shows the time series diagram of chlorophyll-a in Figure 3k. It can be seen that the concentration of chlorophyll-a in SCS and Malacca is significantly higher. In addition, the concentration of chlorophyll-a in samples No. 55-60 was also higher. Chlorophyll-a is a measure of phytoplankton, or algal, biomass (Quinn et al., 2014) and currently most widely used proxy for predicting isoprene concentrations in water (Hackenberg et al., 2017). Numerous studies reported the positive relationship between isoprene emission and chlorophyll-a in the surface seawater (Zhu et al., 2016; Hackenberg et al., 2017). Spatial distributions of marine VOCs are expected to be linked to the distributions of photosynthetic pigments in seawater, such as chlorophyll-a (Ooki et al., 2015). The higher concentrations of chlorophyll-a in the coastal regions stand for higher biological activities and more active to the emission of VOCs (Kang et al., 2018).

8. Minor errors:

L210: it should be "lower than"

**Response:** We have corrected the mistake in the revised manuscript. "The concentrations of total oxoacids are lower than those from Gosan, Jeju Island, South Korea (average 53 ng m$^{-3}$) (Kawamura et al., 2004) and urban sites in China (45 ng m$^{-3}$) (Ho et al., 2007)." (see Page 11, Line 303)

L298-299: should the sentence be "the more the aerosol particles are aged, the higher the ratios are"?

**Response:** We have corrected the mistake in the revised manuscript. "In general, the more the aerosol particles are aged, the higher the $C_2/\Sigma(C_2–C_{12})$ ratios are (Kawamura and Sakaguchi, 1999)." (see Page 14, Line 398–399)

L311: "attribute for" should be "attribute to"

**Response:** We have corrected the mistake in the revised manuscript. (see Page 15, Line 419)

**References:**

Bikkina, S., Kawamura, K., and Miyazaki, Y.: Latitudinal distributions of atmospheric dicarboxylic acids, oxocarboxylic acids, andα-dicarbonyls over the western North Pacific: Sources and formation pathways, Journal of Geophysical Research: Atmospheres, 120, 5010-5035, 10.1002/2014jd022235, 2015.

Hackenberg, S. C., Andrews, S. J., Airs, R., Arnold, S. R., Bouman, H. A., Brewin, R. J. W., Chance, R. J., Cummings, D., Dall'Olmo, G., Lewis, A. C., Minaeian, J. K., Reifel, K. M., Small, A., Tarran, G. A., Tilstone, G. H. and Carpenter, L. J.: Potential controls of isoprene in the surface ocean, Global Biogeochemical Cycles, 31, 644-662, 2017.

Ho, K. F., Cao, J. J., Lee, S. C., Kawamura, K., Zhang, R. J., Chow, J. C., and Watson, J. G.: Dicarboxylic acids, ketocarboxylic acids, and dicarbonyls in the urban atmosphere of China, Journal of Geophysical Research Atmospheres, 112, -, 2007.

Kang M, Fu P, Kawamura K, et al.: Characterization of biogenic primary and secondary organic aerosols in the marine atmosphere over the East China Sea, Atmospheric Chemistry and Physics, 18(19): 13947-13967, 2018.

Kawamura, K., and Gagosian, R. B.: Implications of w-oxocarboxylic acids in the remote marine atmosphere for photo-oxidation of unsaturated fatty acids, Nature, 325, 330-332, 1987.

Kawamura, K., and Sakaguchi, F.: Molecular distribution of water soluble dicarboxylic acids in marine aerosols over the Pacific Ocean including tropics, Journal of Geophysical Research Atmospheres, 104, 3501-3509, 1999.

Kawamura, K., Kobayashi, M., Tsubonuma, N., Mochida, M., Watanabe, T., and Lee, M.: Organic and inorganic compositions of marine aerosols from East Asia: Seasonal variations of water-soluble dicarboxylic acids, major ions, total carbon and nitrogen, and stable C and N isotopic composition, in: The Geochemical Society Special Publications, Elsevier, 243-265, 2004.

Kundu, S., Kawamura, K., and Lee, M.: Seasonal variations of diacids, ketoacids, andα-dicarbonyls in aerosols at Gosan, Jeju Island, South Korea: Implications for sources, formation, and degradation during long-range transport, Journal of Geophysical Research, 115, 10.1029/2010jd013973, 2010.

Kunwar, B., and Kawamura, K.: Seasonal distributions and sources of low molecular weight dicarboxylic acids, ω-oxocarboxylic acids, pyruvic acid, α-dicarbonyls and fatty acids in ambient aerosols from subtropical Okinawa in the western Pacific Rim, Environmental Chemistry, 11, 673-689, 2014.

Kunwar, B., Torii, K., and Kawamura, K.: Springtime influences of Asian outflow and photochemistry on the distributions of diacids, oxoacids and α-dicarbonyls in the aerosols from the western North Pacific Rim, Tellus B: Chemical and Physical Meteorology, 69, 10.1080/16000889.2017.1369341, 2017.

Ooki, A., Nomura, D., Nishino, S., Kikuchi, T. and Yokouchi, Y.: A global-scale map of isoprene and volatile organic iodine in surface seawater of the Arctic, Northwest Pacific, Indian, and Southern Oceans, Journal of Geophysical Research: Oceans, 120, 4108-4128, 2015.

Quinn, P. K., Bates, T. S., Schulz, K. S., Coffman, D. J., Frossard, A. A., Russell, L. M., Keene, W. C. and Kieber, D. J.: Contribution of sea surface carbon pool to organic matter enrichment in sea spray aerosol, Nature Geoscience, 7(3):228-232, 2014.

Zhu, C., Kawamura, K. and Fu, P.: Seasonal variations of biogenic secondary organic aerosol tracers in Cape Hedo, Okinawa, Atmospheric Environment, 130, 113-119, 2016.

---

## Author Comment (AC2) · 8 Mar 2020

**Responses to Reviewer's comments**

We appreciate the reviewer for the thorough reading and thoughtful comments and suggestions, which greatly improve the quality of the manuscript. We have carefully revised the MS accordingly. The point-to-point responses to all the comments are given below in blue.

Reviewer #2 (Formal Review for Author (shown to author)):

The manuscript studied the distributions of dicarboxylic acids, oxocarboxylic acids and α-dicarbonyls in marine aerosols during a cruise from 10 March to 26 April 2015. The cruise area is over South China Sea to East Indian Ocean. There were many samples collected, and the analysis were based on four regions, SCS, EIO-WI, EIO-SL, and SLDP. Through the different concentrations and ratios of dicarboxylic acids, oxoacids and α-dicarbonyls, their sources and possible formation pathways in each studied region were discussed. The work in this manuscript is very important. However, the article still needs to be major revised and then can be published on Atmos. Chem. Phys.

Major Comments:

1. The main meaningful was not very clear during the part of Introduction. The author should give more discussion about this cruise especially the important of this studied area in the Introduction.

   **Response:** Thanks for the reviewer's suggestions. We added the following sentences in the revised manuscript in the section of Introduction.

"Land-sea-air interaction is one of the most important issues in earth system science. Atmospheric aerosol is the major component in the Earth's atmosphere and is one of the key carriers of the global biogeochemical cycle of nutrients (Zhuang et al., 1992;Li et al., 2017;Tan et al., 2011)." (see Page 2, Line 35–37)

"The oceans account for more than 70% of the earth's surface, and marine aerosols are important components of the global aerosol system of natural sources. However, current knowledge about the biogeochemical cycles of organic matters in the tropical marine atmosphere is very limited." (see Page 3, Line 79–81)

"Studies on the molecular composition and distribution of dicarboxylic acids and related compounds can provide useful information for source analysis, secondary formation and photochemical transformation processes of atmospheric organic aerosols." (see Page 3, Line 97–99)

"The South China Sea (SCS) is a large semi-closed marginal basin and one of the largest marginal seas in the world (Liu et al., 2002). The seasonal division of prevailing winds in the South China Sea is mainly influenced by the northeast monsoon roughly from mid-October to mid-March of the following year and by the southwest monsoon from mid-May to mid-September; while from mid-March to mid-May is the spring transition period, during which the wind direction is variable. The climate of the South China Sea is part of the East Asian monsoon system (Lau et al., 1998). The Indian Ocean is the third largest ocean in the world, with distinct tropical maritime and monsoon climate characteristics. The prevailing wind over the Indian Ocean in summer is the southwest monsoon, while the prevailing wind in winter is the northeast monsoon (Fu et al., 2016;Ramanathan et al., 2005). The Indian Ocean is warmer than the Pacific and Atlantic at the same latitude, so it is called the tropical ocean. The tropical East Indian Ocean (10°S–15°N, 65°E–100°E), including the southern bay of Bengal, southeastern Arabian Sea, eastern equatorial Indian Ocean and parts of the southern Indian Ocean, is one of the key regions affecting climate change such as drought and flood in China (Yao et al., 2015)." (see Page 3–4, Line 81–93)

2. There are many analyses and data in the manuscript. It is very difficult for readers to understand the information present in the article because of the illogical.

**Part 3.1.1 Dicarboxylic acids:**

There are many discussions about $C_9$ in paragraph 4, 5, 6, 7, and discussions about Ph in paragraph 5 and 8. The whole part of 3.1.1 is very illogical, and I can't catch the important point and main results. The author should analysis the main connection between these dicarboxylic acids and different areas, give more clearly analysis. For example, form the analysis in the manuscript, the most important dicarboxylic acid is the $C_9$ which have relation with $C_2$–$C_4$, $C_6$, and Ph. The author can put these results together and give the discussion, then give the main point of these results.

**Response:** Thanks for the reviewer's suggestions. The revised manuscript have moved the overall description of dicarboxylic acids and related compounds to the front of Section 3.2.1 (original section 3.1.1), and have reorganized the contents of Section 3.2.1 (original Section 3.1.1) into four paragraphs, i.e. the first paragraph introduces $C_2$–$C_4$, the second paragraph introduces $C_9$, the third paragraph introduces $C_6$, and the fourth paragraph introduces Ph. The logic of the revised manuscript should be clear enough for reader to follow. And at the beginning of Section 3.2.1, a general sentence has been added:

"$C_2$–$C_4$, $C_6$, $C_9$ and Ph are important species of dicarboxylic acids; this section briefly summarizes the main sources and/or formation processes of these organic acids." (see Page 9, Line 238–239)

**Part 3.1.2 Oxocarboxylic acids:**

The meaning of oxocarboxylic acids in the secondary paragraph should be discussed firstly in this part.

**Response:** Thanks for the reviewer's suggestions. At the beginning of Part 3.2.2 (original Part 3.1.2), this manuscript has noted that oxocarboxylic acids mainly contain $\omega C_2$−$\omega C_9$ and pyruvic acid, and have added the following sentence:

"Being similar to diacids, oxocarboxylic acids are mainly derived from combustion sources, but can also be produced by photooxidation of various organic precursors in the atmosphere from anthropogenic and biological sources (Kawamura and Bikkina, 2016)." (see Page 11, Line 297–300)

**Part 3.3.1 Diagnostic mass ratios:**

The data of $C_3/C_4$, $C_2/C_4$, $C_2/\Sigma(C_2-C_{12})$, and M/F can give the information of organic aerosols aging. And they can be put together to give the discussion which can be more clearly.

**Response:** The revised manuscript has moved the content of M/F in Section 3.4 (original Section 3.3.1), and discussed it with $C_3/C_4$, $C_2/C_4$, $C_2/\Sigma(C_2-C_{12})$, to explain the aging of organic aerosols.

**Part 3.3.2 Linear correlations:**

This part can be put in the supporting information, the authors just give the result when other parts need to be supported. For example, line 443-446.

**Response:** This manuscript has merged the content of the original "Section 3.3.1 Diagnostic mass ratios" and the original "Section 3.3.2 Linear correlations", so there is not subtitle in Part 3.3. That is to say, the correlation of $C_2$, $C_3$, and $C_4$, the correlation of $C_2\%$ and $C_2/C_4$, the correlation of Pyr, $\omega C_2$, and MeGly, and the relevant description of chlorophyll a in the original "Section 3.3.2 Linear correlations" has been moved into the "Section 3.3.1 Diagnostic mass ratios", and the rest of original "Section 3.3.2 Linear correlations" has been moved to the supporting information.

3. The authors just compare their data with references, but the discussions are not enough.

Line 158-159, Line 161-164, Line 220 "Oxoacids showed a predominance of ωC2 or ωC3 in five sampling areas (Fig. 3)." Line 221-224.

**Response:** Thanks for the reviewer for pointing this deficiency. Regarding original "Line 158-159", the revised manuscript has added the following discussion after it:

"Therefore, combined with the geographical location of SLDP and the fact that the samples were collected in the ship's docking port, it is speculated that the organic aerosol samples of SLDP may be affected by local coastal and terrestrial biological sources and anthropogenic emissions (especially fossil fuel combustion)." (see Page 8, Line 233–236)

Regarding original "Line 161-164", the revised manuscript has added the following discussion after it: "It can be clearly seen that the concentrations of $C_2$ in SCS and Malacca were higher than those in EIO-SL and EIO-WI (Table 1). The large amount of $C_2$ can be generated from the following sources: fossil fuel combustion (Kawamura and Kaplan, 1987;Donnelly et al., 2010), biomass combustion (Schauer et al., 2001;Narukawa et al., 1999), cooking emissions (Kawamura and Kaplan, 1987;Rogge et al., 1993), photooxidation of VOCs and other precursors (Kawamura and Yasui, 2005;Kundu et al., 2010). The diurnal variation trend of $C_2$ was similar to that of $C_3$ and $C_4$, indicating that these compounds may have similar photochemical oxidation pathways or emission sources in the atmosphere." (see Page 9, Line 242–248)

Regarding original "Line 220-224", the phrase "Oxoacids showed a predominance of ωC$_2$ or ωC$_3$ in five sampling areas (Fig. 3)" has been deleted from the revised manuscript because it overlaps with the next paragraph. And, we have added the following discussion after original "Line 221-224": "The spatial distributions of ω-oxoacids showed a pattern of SLDP > SCS > Malacca > EIO-SL > EIO-WI, being consistent with those of major diacids ($C_2$, $C_3$, and $C_4$), which indicated that these oxoacids were potential precursors of dicarboxylic acids (Sempéré and Kawamura, 1994)." (see Page 11, Line 305–307)

4. There are many repetitions in the article not only the example below.

The sources of $C_9$ and the relation of $C_9$ with other carboxylic acids were discussed repeatedly in part 3.1.1.

Part 3.3.1, the meaning of $C_6$, Ph, MeGly, Gly, Pyr, Isoprene has been given in the former part, delete the repetitions.

**Response:** We have reorganized the contents of Section 3.2.1 (original Section 3.1.1) and merged them into four paragraphs. That is, the first paragraph introduces $C_2$–$C_4$, the second paragraph introduces $C_9$, the third paragraph introduces $C_6$, and the fourth paragraph introduces

Ph. The logic of the revised manuscript is clearer. In addition, we have deleted the meanings of $C_6$, Ph, MeGly, Gly, Pyr and isoprene in Section 3.4 (original Section 3.3.1).

Minor comments:

Line 254 "$C_{10}$" to "$C_{12}$".

  **Response:** Thanks. We have corrected the mistake in the revised manuscript: "Interestingly, the relative abundances of $C_2$ to total mass concentrations of $C_2$ to $C_{12}$ diacids ($\Sigma C_2$–$C_{12}$) were similar in four regions (Fig. 5)." (see Page 12, Line 343–344)

Line 386 delete "isoprene and/or aromatic hydrocarbon oxidation products".

  **Response:** Corrected.

Line 287, "It is worth noting that both C3 and C4 acids show a net loss…"

  **Response:** Thanks. This sentence has been deleted from the revised manuscript because it does not make sense to place it there.

Line 202, delete one "that".

  **Response:** We have corrected the mistake in the revised manuscript: "Temporal variations in $C_9$, $C_6$, and $C_2$−$C_4$ diacids were also similar (Fig. 3), suggesting that photochemical breakdown of $C_9$ might be the major formation pathway of short-chain diacids such as $C_6$, $C_5$, and $C_4$ diacids." (see Page 10, Line 280–282)

**References:**

  Donnelly, T. H., Shergold, J. H., and Southgate, P. N.: Anomalous geochemical signals from phosphatic Middle Cambrian rocks in the southern Georgina Basin, Australia, Sedimentology, 35, 549-570, 2010.
  Fu, P. Q., Aggarwal, S. G., Chen, J., Li, J., Sun, Y. L., Wang, Z. F., Chen, H. S., Liao, H., Ding, A. J., Umarji, G. S., Patil, R. S., Chen, Q., and Kawamura, K.: Molecular Markers of Secondary Organic Aerosol in Mumbai, India, Environmental Science & Technology, 50, 4659-4667, 10.1021/acs.est.6b00372, 2016.
  Kawamura, K., and Kaplan, I. R.: Motor exhaust as a primary source for dicarboxylic acids in Los Angeles ambient air, Environ.sci.technol, 21, 105-110, 1987.
  Kawamura, K., and Yasui, O.: Diurnal changes in the distribution of dicarboxylic acids, ketocarboxylic acids and dicarbonyls in the urban Tokyo atmosphere, Atmospheric Environment, 39, 1945-1960, 2005.
  Kundu, S., Kawamura, K., and Lee, M.: Seasonal variations of diacids, ketoacids, andα-dicarbonyls in aerosols at Gosan, Jeju Island, South Korea: Implications for sources,

formation, and degradation during long-range transport, Journal of Geophysical Research, 115, 10.1029/2010jd013973, 2010.

Lau, K.-M., Wu, H.-T., and Yang, S.: Hydrologic processes associated with the first transition of the Asian summer monsoon: A pilot satellite study, Bulletin of the American Meteorological Society, 79, 1871-1882, 1998.

Li, W. J., Xu, L., Liu, X. H., Zhang, J. C., Lin, Y. T., Yao, X. H., Gao, H. W., Zhang, D. Z., Chen, J. M., Wang, W. X., Harrison, R. M., Zhang, X. Y., Shao, L. Y., Fu, P. Q., Nenes, A., and Shi, Z. B.: Air pollution-aerosol interactions produce more bioavailable iron for ocean ecosystems, Science Advances, 3, 10.1126/sciadv.1601749, 2017.

Liu, K. K., Chao, S. Y., Shaw, P. T., Gong, G. C., Chen, C. C., and Tang, T. Y.: Monsoon-forced chlorophyll distribution and primary production in the South China Sea: observations and a numerical study, Deep Sea Research Part I Oceanographic Research Papers, 49, 1387-1412, 2002.

Narukawa, M., Kawamura, K., Takeuchi, N., and Nakajima, T.: Distribution of dicarboxylic acids and carbon isotopic compositions in aerosols from 1997 Indonesian forest fires, Geophysical Research Letters, 26, 3101-3104, 1999.

Ramanathan, V., Chung, C., Kim, D., Bettge, T., Buja, L., Kiehl, J. T., Washington, W. M., Fu, Q., Sikka, D. R., and Wild, M.: Atmospheric brown clouds: Impacts on South Asia climate and hydrological cycle, PNAS, 102, 5326-5333, 2005.

Rogge, W. F., Mazurek, M. A., Hildemann, L. M., Cass, G. R., and Simoneit, B. R.: Quantification of urban organic aerosols at a molecular level: identification, abundance and seasonal variation, Atmospheric Environment. Part A. General Topics, 27, 1309-1330, 1993.

Schauer, J. J., Kleeman, M. J., Cass, G. R., and Simoneit, B. R.: Measurement of emissions from air pollution sources. 3. C1-C29 organic compounds from fireplace combustion of wood, Environmental Science & Technology, 35, 1716-1728, 2001.

Sempéré, R., and Kawamura, K.: Comparative Distributions of Dicarboxylic-Acids and Related Polar Compounds in Snow Rain and Aerosols from Urban Atmosphere, Atmos. Environ., 28, 449-459, 1994.

Tan, S. C., Shi, G. Y., Shi, J. H., Gao, H. W., and Yao, X. H.: Correlation of Asian dust with chlorophyll and primary productivity in the coastal seas of China during the period from 1998 to 2008, Journal of Geophysical Research-Biogeosciences, 116, 10.1029/2010jg001456, 2011.

Yao, T., Wu, F., Ding, L., Sun, J., Zhu, L., Piao, S., Deng, T., Ni, X., Zheng, H., and Ouyang, H.: Multispherical interactions and their effects on the Tibetan Plateau's earth system: a review of the recent researches, Nat. Sci. Rev., 2, 468-488, 2015.

Zhuang, G. S., Yi, Z., Duce, R. A., and Brown, P. R.: Link between Iron and Sulfur Cycles Suggested by Detection of Fe(II) in Remote Marine Aerosols, Nature, 355, 537-539, 1992.

---

## Author Response (AR2)

**Responses to the Reviewer's Comments**

We appreciate the reviewer for the thorough reading and thoughtful comments and suggestions, which greatly improve the quality of the manuscript. We revised the MS accordingly. The point-to-point responses to all the comments are given below in blue.

Suggestions/comments:

1. Title: the title could be modified a bit. What does "molecular distribution" mean? "over the South China Sea and East Indian Ocean" could be changed to "from the South China Sea to the East Indian Ocean".

**Response:** Molecular distribution means the distribution characteristics of the concentrations of various organic acids with different carbon numbers at different time and places. And, following the reviewer's suggestion, we have changed the title to "Molecular and spatial distributions of dicarboxylic acids, oxocarboxylic acids and α-dicarbonyls in marine aerosols from the South China Sea to East Indian Ocean".

2. Table 3 could be moved to the supporting information.

**Response:** In the revision, we have moved Table 3 to the supporting information (SI).

3. The concentrations in the SLDP should be taken for further consideration. As mentioned in the sampling part, these samples mainly show the severe influence of ship exhaust. They are not ambient aerosol samples if compared to other samples.

**Response:** The aerosol samples of SLDP were affected by multiple sources, namely local coastal and terrestrial biological sources and anthropogenic emissions (especially fossil fuel combustion). In addition, the 5-day backward trajectory showed that the air mass originated from the land of Sri Lanka and the surrounding sea area, which indicated that the samples were not only polluted by ship exhaust. Thus, the SLDP samples were ambient aerosol samples. The reasons for the analysis of SLDP samples in this paper are as follows: firstly, it can be roughly compared with aerosol samples derived from SCS and EIO; secondly, it is meaningful and interesting to study the samples affected by ship exhaust pollution along the coast. Therefore, it is necessary to study and analyze SLDP samples in this paper.

4. The comparison should be more logical in the section 3.1 (L156-173), 3.2 (L217-232) and other parts. It is not easy to follow these parts. For what reason the authors would like to make the comparison if it is really necessary. E.g. The concentrations of OC and EC in the SCS could

be further compared with many other recent observations in this area. Why compared with those from Okinawa (P160-166)?

**Response:** In the section 3.1 (L156-173), we have compared the results of this study with those of other sea areas that were more affected by terrestrial long-distance transmission and those that were less affected by terrestrial anthropogenic emissions. We have added comparisons with some other recent observations in this area and its vicinity. The revised content is as follows.

"The mass concentrations of OC and EC in SCS were lower than those in the western South China Sea (OC: 2.04 μg m$^{-3}$, EC: 3.0 μg m$^{-3}$) which were strongly affected by biomass burning (Song et al., 2018). In addition, the carbon component concentration levels in Malacca were similar to those in the Bay of Bengal (OC: 1.9 μg m$^{-3}$, EC: 0.4 μg m$^{-3}$) (Kumar et al., 2008)." (see Page 6, Line 165–168).

"The concentration levels of carbon components in EIO-WI and EIO-SL were lower than those in the Indian Ocean (OC: 0.71 μg m$^{-3}$, EC: 0.49 μg m$^{-3}$) (Fu et al., 2011)." (see Page 7, Line 175–176).

In the section 3.2 (L217–232), we want to compare the results of this study with other marine regions, polar regions, and urban regions. In order to make the comparison more logical, the contents are modified as follows.

"Table 1 shows the total concentrations of diacids in aerosol samples from five sea areas in this study (SCS: 0.27±0.15 ng m$^{-3}$; EIO-WI: 0.05±0.05 ng m$^{-3}$; EIO-SL: 0.04±0.18 ng m$^{-3}$; Malacca: 0.24±0.18 ng m$^{-3}$; and SLDP: 0.27±0.15 ng m$^{-3}$). Fu et al. (2013) reported that the mean total diacids concentration of aerosol samples in the South China Sea and the Indian Ocean with the sampling period of three days was 489 ng m$^{-3}$ and 301 ng m$^{-3}$, respectively. Comparing this study with aerosols from other marine areas, it was found that the total diacids concentration of aerosol samples in the EIO was similar to that in the western Pacific Ocean (60±39 ng m$^{-3}$) (Wang et al., 2006). What's more, the concentration of total diacids in aerosol samples from five sea areas was all lower than that of the East China Sea (850 ng m$^{-3}$) (Mochida et al., 2003), and higher than that of the Southern Ocean (4.5±4.0 ng m$^{-3}$) (Wang et al., 2006). In addition, the concentration of total diacids in aerosol samples in all sea areas was higher than that in the Alert, Arctic (25 ng m$^{-3}$) (Kawamura et al., 1996) in the polar region, and lower than that in the urban center of Xi'an, China in different dust events (932–2240 ng m$^{-3}$) (Wang et al., 2015), the megacity Chennai, India (694.5±176.3 and 640.6±150.6 ng m$^{-3}$ in early and late winter) (Pavuluri et al., 2010), and the megacity Chennai, India in summer (502.9±117.9 ng m$^{-3}$) (Pavuluri et al., 2010) and other urban areas." (see Page 8, Line 220–232).

5. L216: what do the similar formation processes mean? The sources of diacids in the coastal and in the remote regions could be very different if you read through these and other related references.

    **Response:** The revised content is as follows.

    "The dominance of $C_2$ followed by $C_3$ and $C_4$ diacids is consistent with those coastal marine aerosols (Kundu et al., 2010;Kunwar and Kawamura, 2014) and remote marine aerosols from the western North Pacific (Bikkina et al., 2015), suggesting similar formation processes of dicarboxylic acids in the atmosphere of the coastal or remote regions." (see Page 8, Line 216–219)

6. L286-287: Again, why compare with the concentrations in urban Tokyo?

    **Response:** The revised content is as follows.

    "Concentrations of Ph acid in EIO-WI and EIO-SL were similar to those of the western North Pacific (MBA: 0.6 ng m$^{-3}$; LBA: 0.5 ng m$^{-3}$) (Bikkina et al., 2015), where were marine-derived contributions. In addition, concentrations of Ph acid over the SCS to EIO were much lower than those reported in the urban atmosphere such as Chennai (21 ng m$^{-3}$) (Pavuluri et al., 2010) and Chinese cities (90 ng m$^{-3}$) (Ho et al., 2007)." (see Page 10, Line 287–291)

7. L291-294: How about the influence of ship exhaust on the increase of Ph?

    **Response:** Ship exhaust can increase the atmospheric level of phthalic acid (Ph). Phthalic acid is either formed via photochemical pathways of naphthalene or directly released into air by fossil fuel burning and the incomplete combustion of aromatic hydrocarbons in motor vehicles. Moreover, the abundance of Ph may also be caused by increased phthalates emissions from plastic waste burnings in heavily polluted areas in China (Deshmukh et al., 2016). In this study, Ph was more abundant in SLDP by several folds than other regions. Combined with the geographical location of SLDP and the fact that the samples were collected in the ship's docking port, it is speculated that the organic aerosol samples of SLDP may be affected by local coastal and terrestrial biological sources and anthropogenic emissions (especially fossil fuel combustion). In addition, EC is the product of residential coal, fossil fuel combustion and incomplete combustion of biomass. The concentration of EC in SLDP aerosol (4.24±3.02 μg m$^{-3}$) was also much higher than that in other marine areas (SCS: 0.27±0.15 μg m$^{-3}$, EIO-WI: 0.05±0.05 μg m$^{-3}$, EIO-SL: 0.21±0.18 μg m$^{-3}$, Malacca: 0.24±0.18 μg m$^{-3}$), indicating that the aerosol samples of SLDP were seriously polluted by ship exhaust but not for the other marine samples. EC is the product of residential coal, fossil fuel combustion and incomplete

combustion of biomass. The concentration of EC in SLDP aerosol ($4.24\pm3.02$ µg m$^{-3}$) was also much higher than that in other marine areas (SCS: $0.27\pm0.15$ µg m$^{-3}$, EIO-WI: $0.05\pm0.05$ µg m$^{-3}$, EIO-SL: $0.21\pm0.18$ µg m$^{-3}$, Malacca: $0.24\pm0.18$ µg m$^{-3}$), indicating that the aerosol samples of SLDP were seriously polluted by ship exhaust but not for the other marine samples.

8. Section 3.5 could be merged in Section 3.2.

   **Response:** Section 3.5 has been merged into Section 3.2 and becomes Section 3.2.4.

[revised manuscript text omitted]